

# Symmetries and anomalies of Kitaev spin-S models: Identifying symmetry-enforced exotic quantum matter

**Ruizhi Liu[1,2], Ho Tat Lam[3], Han Ma[2] and Liujun Zou[2]**

**1** Department of Mathematics and Statistics, Dalhousie University,
Halifax, Nova Scotia, Canada, B3H 4R2
**2** Perimeter Institute for Theoretical Physics, Waterloo, Ontario, Canada N2L 2Y5
**3** Department of Physics, Massachusetts Institute of Technology,
Cambridge, Massachusetts 02139, USA

## Abstract

We analyze the internal symmetries and their anomalies in the Kitaev spin-$S$ models. Importantly, these models have a lattice version of a $\mathbb{Z}_2$ 1-form symmetry, denoted by $\mathbb{Z}_2^{[1]}$. There is also an ordinary 0-form $\mathbb{Z}_2^{(x)} \times \mathbb{Z}_2^{(y)} \times \mathbb{Z}_2^T$ symmetry, where $\mathbb{Z}_2^{(x)} \times \mathbb{Z}_2^{(y)}$ are $\pi$ spin rotations around two orthogonal axes, and $\mathbb{Z}_2^T$ is the time reversal symmetry. The anomalies associated with the full $\mathbb{Z}_2^{(x)} \times \mathbb{Z}_2^{(y)} \times \mathbb{Z}_2^T \times \mathbb{Z}_2^{[1]}$ symmetry are classified by $\mathbb{Z}_2^{17}$. We find that for $S \in \mathbb{Z}$ the model is anomaly-free, while for $S \in \mathbb{Z} + \frac{1}{2}$ there is an anomaly purely associated with the 1-form symmetry, but there is no anomaly purely associated with the ordinary symmetry or mixed anomaly between the 0-form and 1-form symmetries. The consequences of these symmetries and anomalies apply to not only the Kitaev spin-$S$ models, but also any of their perturbed versions, assuming that the perturbations are local and respect the symmetries. If these local perturbations are weak, generically these consequences still apply even if the perturbations break the 1-form symmetry. A notable consequence is that there should generically be a deconfined fermionic excitation carrying no fractional quantum number under the $\mathbb{Z}_2^{(x)} \times \mathbb{Z}_2^{(y)} \times \mathbb{Z}_2^T$ symmetry if $S \in \mathbb{Z} + \frac{1}{2}$, which implies symmetry-enforced exotic quantum matter. We also discuss the consequences for $S \in \mathbb{Z}$.



# 1 Introduction

A central goal of condensed matter physics is to realize interesting quantum phases of matter. Kitaev's solvable spin-1/2 model [1] provides a theoretical foundation for various quantum spin liquid phases, such as Abelian or non-Abelian topological orders, and gapless quantum spin liquids. It also attracts tremendous attention due to the discovery of candidate materials [2–5]. More recently, the higher-spin generalizations of the Kitaev materials were proposed [6–12] and have triggered extensive analytical and numerical studies [13–28].

Theoretically, it is important to first understand the higher-spin generalizations of the Kitaev spin-1/2 model. Unlike the spin-1/2 model, no analytic solution is known for the Kitaev spin-$S$ models, for $S > 1/2$. However, by using a carefully designed parton construction, one of us proved the presence of an exact $\mathbb{Z}_2$ gauge structure in this model. Moreover, there is an even-odd effect: If $S \in \mathbb{Z} + \frac{1}{2}$ then the gauge charge is fermionic and the $\mathbb{Z}_2$ gauge field is deconfined, while if $S \in \mathbb{Z}$ the gauge charge is bosonic and the $\mathbb{Z}_2$ gauge field can be Higgsed [28].

The observations in Ref. [28] raise some fundamental questions. First, that work focuses on the Kitaev spin-$S$ Hamiltonian, so a pertinent question is: To what extent is the even-odd effect stable against perturbation, which can potentially be strong? Second, Ref. [28] employs a parton construction to unveil the gauge structure. So another question is: Can the results therein be obtained via a more elementary, direct approach, without searching for an exact parton construction? A third question is: Can one obtain more results than those in Ref. [28]? For example, that work predicts a deconfined fermionic excitation for $S \in \mathbb{Z} + \frac{1}{2}$, and a natural question is: Can this fermion carry fractional quantum numbers under the symmetries of the model? Similar questions were also raised in the Journal Club for Condensed Matter Physics [29].



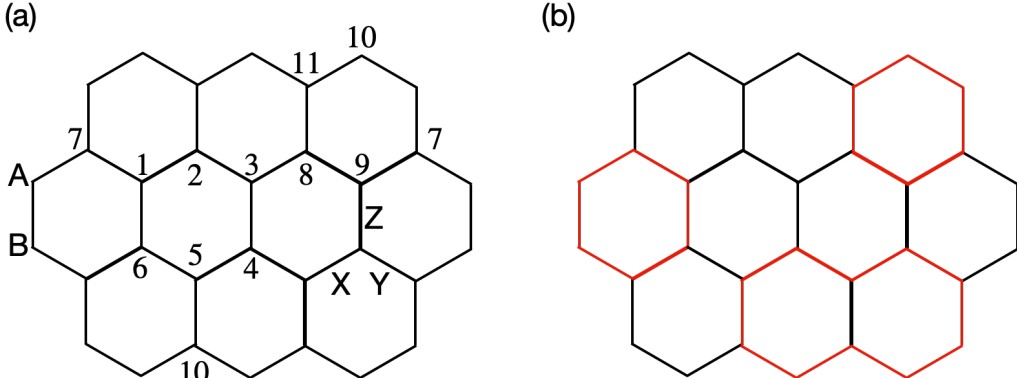

Figure 1: (a) A honeycomb lattice with periodic boundary conditions along the two directions. $X$, $Y$ and $Z$ label the bonds, $A$ and $B$ label the sublattices, and numbers label the sites. Sites labeled by the same number are identified. (b) Examples of closed loop operators, whose supports are in red.

In this paper, we address the above questions by studying the symmetries and anomalies of the Kitaev spin-$S$ models. Besides some ordinary 0-form symmetries, we find that the Kitaev spin-$S$ models have an exact 1-form symmetry [30] (see Refs. [31, 32] for reviews on 1-form symmetries). Moreover, this 1-form symmetry is anomalous if $S \in \mathbb{Z} + \frac{1}{2}$, while it is anomaly-free if $S \in \mathbb{Z}$. We also show that, for all $S$, there is no anomaly solely associated with the ordinary 0-form symmetries or mixed anomaly between the 0-form and 1-form symmetries.

These results sharpen and generalize the observations in Ref. [28]. We will discuss the profound consequences of these symmetries and anomalies in detail in Sec. 5. For now, we remark that our arguments apply to generic local Hamiltonian that respects the relevant symmetries, which can deviate significantly from the Kitaev spin-$S$ models. Moreover, even if the 1-form symmetry is slightly broken by local perturbations, these consequences are still robust. One consequence that should be highlighted here is that if $S \in \mathbb{Z} + \frac{1}{2}$, then the system should generically host deconfined fermionic excitations, which means that this system realizes symmetry-enforced exotic quantum matter, where the symmetry enforces the system to be a quantum spin liquid.

## 2 Kitaev spin-$S$ model and its symmetries

The Kitaev model is defined for spin-$S$ moments living at the sites of a honeycomb lattice (see Fig. 1), with the Hamiltonian

$$H = -\sum_{\mu} J_{\mu} \sum_{\langle i,j \rangle \in \mu} S_i^{\mu} S_j^{\mu}, \tag{1}$$

where $\mu = x, y, z$ labels the bonds, and $\langle i, j \rangle \in \mu$ represents the two sites connected by a bond $\mu$. For generic values of $J_{\mu}$, this model enjoys many symmetries:

1. Lattice version of $\mathbb{Z}_2$ 1-form symmetry, denoted by $\mathbb{Z}_2^{[1]}$. For each plaquette, there is a generator of this symmetry, such as[1]

$$W_p = e^{i\pi S_1^y} e^{i\pi S_2^z} e^{i\pi S_3^x} e^{i\pi S_4^y} e^{i\pi S_5^z} e^{i\pi S_6^x}, \tag{2}$$

---

[1]Depending on the eigenvalue of the ground state under $W_p$, there can be an additional $-1$ prefactor in the definition of the generator. But this prefactor will not affect our discussion and will be ignored below (see Appendix A for more details).

where the subscripts are site labels (see Fig. 1(a)) [1, 13]. If the system is on a torus (i.e., under periodic boundary conditions), there is one more generator along each non-contractible cycle of the torus. In Fig. 1(a), these generators are

$$W_1 = e^{i\pi S_7^z}e^{i\pi S_1^z}e^{i\pi S_2^z}e^{i\pi S_3^z}e^{i\pi S_8^z}e^{i\pi S_9^z},$$
$$W_2 = e^{i\pi S_{10}^y}e^{i\pi S_5^y}e^{i\pi S_4^y}e^{i\pi S_3^y}e^{i\pi S_8^y}e^{i\pi S_{11}^y}.$$

$$(3)$$

All these generators commute. The presence of this symmetry is often phrased as the conservation of $W_p$, but we regard it as a lattice version of a 1-form symmetry, because its generators and their products are supported on all possible closed loops (see Fig. 1(b)), just as a 1-form symmetry in a $2+1$ dimensional continuum field theory [30]. Moreover, since $W_p^2 = 1$, this symmetry should be regarded as a $\mathbb{Z}_2$ symmetry. In Appendix A, we present more discussion on the lattice version of a general $\mathbb{Z}_n$ 1-form symmetry.

2. $\mathbb{Z}_2^T$ anti-unitary time reversal symmetry, whose action on each spin operator is $S_i^\mu \to -S_i^\mu$.

3. $\mathbb{Z}_2^{(x)} \times \mathbb{Z}_2^{(y)}$, generated by $\pi$ spin rotations around $S^x$ and $S^y$. Namely, the generators of $\mathbb{Z}_2^{(x)}$ and $\mathbb{Z}_2^{(y)}$ are $\prod_i e^{i\pi S_i^x}$ and $\prod_i e^{i\pi S_i^y}$, respectively.

There is also a lattice translation symmetry and a 2-fold lattice rotation symmetry, but in this paper we will focus on the above $\mathbb{Z}_2^{[1]} \times \mathbb{Z}_2^{(x)} \times \mathbb{Z}_2^{(y)} \times \mathbb{Z}_2^T$ internal symmetry. In the isotropic limit where $J_x = J_y = J_z$, there are additional 3-fold lattice rotation symmetry and reflection symmetry. We leave a systematic study of the effects of lattice symmetries to future work.

## 3  1-form symmetry and its anomaly

The $\mathbb{Z}_2^{[1]}$ symmetry is a notable feature of the Kitaev spin-$S$ model, Eq. (1). Now we show that this symmetry is anomalous (non-anomalous) for all half-odd-integer spins (integer spins), i.e., $S \in \mathbb{Z} + \frac{1}{2}$ ($S \in \mathbb{Z}$). For certain specific values of $S$ (e.g., 1/2), this anomaly was discussed in Refs. [33–35].

### 3.1  Statistics of the end point excitations

To understand the anomaly of the 1-form symmetry, recall that an anomaly is an obstruction to gauging this symmetry, i.e., coupling the system to a gauge field for this symmetry. We will illustrate this obstruction in Sec. 3.2, and here we use a simpler and more illuminating method to detect this anomaly. Specifically, we will cut open the loops on which the 1-form symmetry generators are supported (e.g., the red loops in Fig. 1 (b)). Because the loop operators are symmetries and the Hamiltonian is local, the resulting open string operators are tensionless and they create deconfined point-like excitations around their end points (unless these excitations are condensed). Next, we check the statistics of the end points of these open strings. It is known that a 1-form symmetry is anomalous unless this statistics is bosonic, because gauging a 1-form symmetry can be viewed as condensing these end points, but they can condense only if they are bosons [36,37].

To check the statistics of these end points, we use the approach in Refs. [38, 39]. This approach was originally designed for topologically ordered ground states, but we do not need any assumption about the ground state. The general idea is illustrated in Fig. 2(a). Unitary operators $M_{1,2,3}$ in Fig. 2(a) can freely move the end points. Suppose we apply $M_1M_2M_3$ to a state with two end points, and denote the final state by $|1\rangle$. We can also apply $M_3M_2M_1$ to the same initial state, and denote the final state by $|2\rangle$. Comparing states $|1\rangle$ and $|2\rangle$, we see that

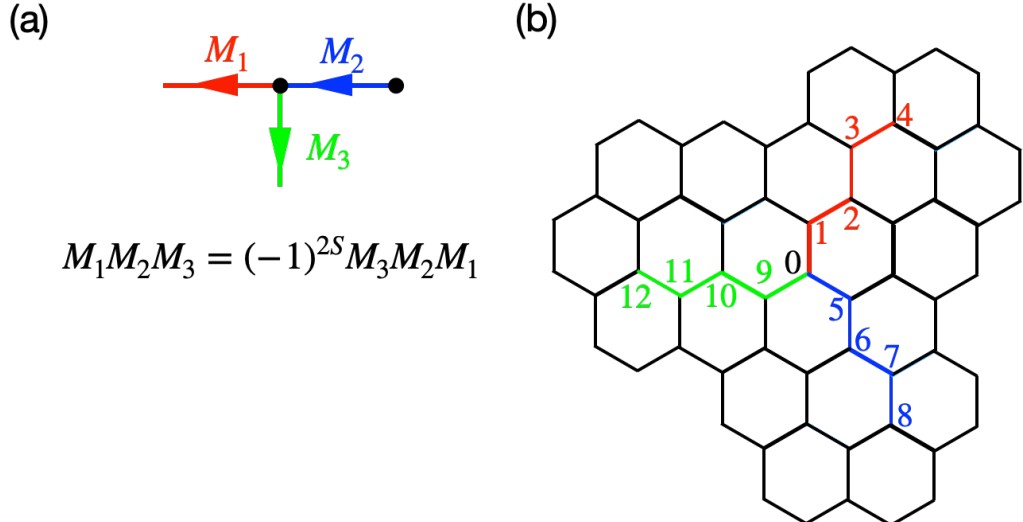

Figure 2: Determination of the statistics of the end points. (a) The two black points represent two end points, and $M_1$, $M_2$ and $M_3$ are operators that move the end points. (b) The red, blue and green strings represent the support of $M_1$, $M_2$ and $M_3$, respectively. The numbers label the sites.

they differ by a position exchange of the two end points. So the relative phase factor between these two states, $e^{i\theta}$, gives the statistics of the end points. These sequences of operations are carefully chosen so that in $e^{i\theta}$ all non-universal details are canceled, and only the universal information of the statistics is kept.

Now we apply this approach to the Kitaev spin-$S$ model. Suppose initially there are two end points at sites 0 and 8 in Fig. 2(b). The operators $M_{1,2,3}$ can be chosen as

$$
\begin{aligned}
M_1 &= U_4 e^{i\pi(S_3^y + S_2^y + S_1^y)} U_0^{(1)}, \\
M_2 &= U_0^{(2)} e^{i\pi(S_5^x + S_6^x + S_7^x)} U_8, \\
M_3 &= U_{12} e^{i\pi(S_{11}^z + S_{10}^z + S_9^z)} U_0^{(3)},
\end{aligned}
\tag{4}
$$

where the subscript of each operator is its site index. To ensure the absence of energy cost when moving the end points, these operators are chosen so that in the interior of each string the operators are simply the part of a closed loop operator in this region, but at the end points of the strings we can put more general unitary operators, such as $U_0^{(1,2,3)}$. In order for these strings to seamlessly connect to become longer strings, we demand

$$
U_0^{(1)} U_0^{(2)} = \lambda_1 e^{i\pi S_0^x}, \qquad U_0^{(3)} U_0^{(2)} = \lambda_2 e^{i\pi S_0^z},
\tag{5}
$$

where $\lambda_{1,2}$ are some phase factors.

Now we show $M_1 M_2 M_3 = (-1)^{2S} M_3 M_2 M_1$. To this end, it suffices to show that $U_0^{(1)} U_0^{(2)} U_0^{(3)} = (-1)^{2S} U_0^{(3)} U_0^{(2)} U_0^{(1)}$, which simply follows from Eq. (5) (using $e^{i\pi S_0^x} e^{i\pi S_0^z} = (-1)^{2S} e^{i\pi S_0^z} e^{i\pi S_0^x}$). One can check that if we deform the shapes of the strings or change the positions of their end points, as long as the strings connect in a way as in Fig. 2(a), the relation $M_1 M_2 M_3 = (-1)^{2S} M_3 M_2 M_1$ always holds. Namely, the statistics phase $e^{i\theta} = (-1)^{2S}$.

Therefore, we conclude that the end points of the strings related to the $\mathbb{Z}_2^{[1]}$ symmetry have fermionic (bosonic) statistics if $S \in \mathbb{Z} + \frac{1}{2}$ ($S \in \mathbb{Z}$), so this symmetry is anomalous (non-anomalous).

In passing, we note this anomaly can also be seen from the anisotropic limit of Eq. (1), where $|J_x| \gg |J_{y,z}|$. In this limit, the model realizes a $\mathbb{Z}_2$ topological order if $S \in \mathbb{Z} + \frac{1}{2}$, such that the closed loop operators are precisely the Wilson loops of some fermionic excitations. If $S \in \mathbb{Z}$, the model realizes a short-range entangled ground state [28]. These again imply that the $\mathbb{Z}_2^{[1]}$ symmetry is anomalous (non-anomalous) if $S \in \mathbb{Z} + \frac{1}{2}$ ($S \in \mathbb{Z}$). The advantage of the method illustrated in Fig. 2 is its generality, since it only uses the symmetry properties and does not rely on any solvable limit of any Hamiltonian.

## 3.2 Gauging the $\mathbb{Z}_2^{[1]}$ symmetry

Now we gauge the $\mathbb{Z}_2^{[1]}$ symmetry in the Kitaev spin-$S$ model, and we will see an obstruction to this gauging if $S \in \mathbb{Z} + \frac{1}{2}$, while this obstruction does not show up if $S \in \mathbb{Z}$, which confirms the presence (absence) of the $\mathbb{Z}_2^{[1]}$ symmetry anomaly for $S \in \mathbb{Z} + \frac{1}{2}$ ($S \in \mathbb{Z}$). We remark that the $\mathbb{Z}_2^{[1]}$ symmetry action here appears to be "on-site", but this obstruction to gauging still exists for $S \in \mathbb{Z} + \frac{1}{2}$. This is in sharp contrast to 0-form unitary symmetries, which can always be gauged if their actions are on-site in a tensor-product Hilbert space. Readers less familiar with gauging on lattices can skip this subsection for the first reading.

Given a generator defined on a loop $\gamma$ on the lattice, the global $\mathbb{Z}_2^{[1]}$ symmetry action is

$$S_i^\mu \to \lambda_i^\mu S_i^\mu \,, \tag{6}$$

where

$$\lambda_i^\mu = \begin{cases} -1 \,, & i, \mu \in \gamma \,, \\ 1 \,, & \text{otherwise.} \end{cases} \tag{7}$$

Here $i, \mu \in \gamma$ means that the site $i$ is on $\gamma$ and the $\mu$-bond adjacent to the site $i$ is also on $\gamma$. The above condition should be thought as a lattice version of the closedness condition in a gauge theory, $\delta\lambda = 0$.

To couple the system to a background field of the $\mathbb{Z}_2^{[1]}$ symmetry, we remove the closedness condition Eq. (7). So the gauge transformation for spins are

$$S_i^\mu \to \lambda_i^\mu S_i^\mu \,, \quad \lambda_i^\mu = \pm 1 \,. \tag{8}$$

The gauge invariance of the gauged Hamiltonian requires us to include a gauge field $A_{ij}$ defined on each link, with gauge transformation

$$A_{ij} \to \lambda_i^\mu A_{ij} \lambda_j^\mu \,, \quad \langle i, j \rangle = \mu \,. \tag{9}$$

The gauge field on each link can be viewed as a two-state system, and $A$ can be represented by the Pauli operator $\sigma_3$. The minimally coupled Hamiltonian is then

$$H' = -\sum_\mu J_\mu \sum_{\langle i,j \rangle = \mu} S_i^\mu A_{ij} S_j^\mu \,. \tag{10}$$

Below we will show that the above gauging procedure is actually problematic if $S \in \mathbb{Z} + \frac{1}{2}$, by showing that the Gauss law constraints cannot be simultaneously satisfied. On the other hand, the above gauging procedure is valid if $S \in \mathbb{Z}$.

Given a lattice site $k$, it belongs to 3 different hexagons in the honeycomb lattice. There are 3 Gauss law constraints, but only 2 of them are independent. These Gauss laws are related to the following operators that generate the gauge transformations in Eqs. (8) and (9):

$$\begin{aligned} G_k^{(1)} &= \exp(i\pi S_k^x) E_k^y E_k^z \,, \\ G_k^{(2)} &= \exp(i\pi S_k^y) E_k^x E_k^z \,, \\ G_k^{(3)} &= G_k^{(1)} G_k^{(2)} \,, \end{aligned} \tag{11}$$

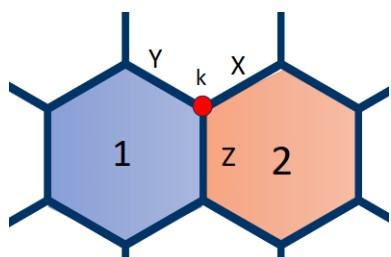

Figure 3: Independent Gauss law constraints from hexagons 1 and 2.

where $E$ is the conjugate momentum of $A$ defined on each link, i.e., $EAE^{-1} = -A$. For example, $E_k^x$ means the conjugate variable on $x$-link adjacent to site $k$. If the gauge field on a link is represented as a two-state system, then $E$ can be represented by the Pauli operator $\sigma_1$. The superscripts in $G_k^{(1,2,3)}$ label which hexagon we are working on. It is easy to check that Eq. (11) generates the correct gauge transformation, given by Eqs. (8) and (9).

After gauging, the physical Hilbert space is spanned by gauge invariant states $|\text{phy}\rangle$ satisfying

$$G_k^{(i)}|\text{phy}\rangle = |\text{phy}\rangle . \tag{12}$$

Notably, the Gauss law operators obey an algebra depending on the spin $S$

$$G_k^{(1)}G_k^{(2)} = (-1)^{2S} G_k^{(2)}G_k^{(1)} . \tag{13}$$

When $S \in \mathbb{Z}$, these operators commute, so they can be simultaneously diagonalized and the physical Hilbert space is non-empty. In contrast, when $S \in \mathbb{Z} + \frac{1}{2}$, these Gauss law operators do not commute so the Gauss law constraints Eq. (12) cannot be simultaneously satisfied, i.e., the physical Hilbert space is actually empty. This signifies the obstruction to gauging the 1-form symmetry when $S \in \mathbb{Z} + \frac{1}{2}$, i.e., the anomaly associated with the $\mathbb{Z}_2^{[1]}$ symmetry. Such an obstruction exists although the action of $\mathbb{Z}_2^{[1]}$ appears to be "on-site".

# 4 Full internal symmetry anomaly

After identifying the anomaly associated with the $\mathbb{Z}_2^{[1]}$ symmetry, in this section we discuss the full anomaly associated with the internal $\mathbb{Z}_2^{(x)} \times \mathbb{Z}_2^{(y)} \times \mathbb{Z}_2^T \times \mathbb{Z}_2^{[1]}$ symmetry. We will see that for all $S$, there is no anomaly purely associated with the 0-form $\mathbb{Z}_2^{(x)} \times \mathbb{Z}_2^{(y)} \times \mathbb{Z}_2^T$ symmetry or mixed anomaly between the 0-form $\mathbb{Z}_2^{(x)} \times \mathbb{Z}_2^{(y)} \times \mathbb{Z}_2^T$ symmetry and the $\mathbb{Z}_2^{[1]}$ symmetry, and all anomalies are purely associated with the $\mathbb{Z}_2^{[1]}$ symmetry, which is discussed in Sec. 3.

The classification of all anomalies associated with the $\mathbb{Z}_2^{(x)} \times \mathbb{Z}_2^{(y)} \times \mathbb{Z}_2^T \times \mathbb{Z}_2^{[1]}$ symmetry is $\mathbb{Z}_2^{17}$, with the details given in Appendix B. Here we give a more physics-oriented explanation of this classification. It turns out that for this purpose it is convenient to view $\mathbb{Z}_2^{17} = \mathbb{Z}_2^{10} \times \mathbb{Z}_2 \times \mathbb{Z}_2^6$, then each piece has a simple interpretation.

- $\mathbb{Z}_2^{10}$ piece, containing anomalies solely associated with the 0-form $\mathbb{Z}_2^{(x)} \times \mathbb{Z}_2^{(y)} \times \mathbb{Z}_2^T$ symmetry, which have been classified in the condensed matter literature. The group cohomology theory gives a $H^4(\mathbb{Z}_2^{(x)} \times \mathbb{Z}_2^{(y)} \times \mathbb{Z}_2^T, U(1)) = \mathbb{Z}_2^9$ classification [40], and there is another "beyond-cohomology" anomaly [41], which together give $\mathbb{Z}_2^{10}$.

- $\mathbb{Z}_2$ piece, containing anomalies solely associated with the $\mathbb{Z}_2^{[1]}$ 1-form symmetry. As mentioned before, such anomalies are simply classified by $e^{i\theta}$, the statistics of the end

points of the strings related to this symmetry. In general, this statistics can be anyonic. But time reversal symmetry requires $e^{i\theta} = e^{-i\theta}$, or $e^{i\theta} = \pm 1$, which means this statistics is either bosonic or fermionic, and it contributes a $\mathbb{Z}_2$ classification to the full anomaly.

- $\mathbb{Z}_2^6$ piece, containing mixed anomalies between the 0-form and 1-form symmetries. As explained in Appendix C (see also Refs. [42, 43]), these anomalies are in one-to-one correspondence with different fractionalization patterns of the $\mathbb{Z}_2^{(x)} \times \mathbb{Z}_2^{(y)} \times \mathbb{Z}_2^T$ symmetry on the end points of the strings associated with the $\mathbb{Z}_2^{[1]}$ symmetry. Indeed, these patterns are classified by $H^2(\mathbb{Z}_2^{(x)} \times \mathbb{Z}_2^{(y)} \times \mathbb{Z}_2^T, \mathbb{Z}_2) = \mathbb{Z}_2^6$ [44], and they can be organized as follows.

  1. The end points carry half charge under $\mathbb{Z}_2^{(x)}$.

  2. The end points carry half charge under $\mathbb{Z}_2^{(y)}$.

  3. $\mathbb{Z}_2^{(x)}$ and $\mathbb{Z}_2^{(y)}$ anti-commute when they act on the end points.

  4. The end points are Kramers doublets.

  5. $\mathbb{Z}_2^{(x)}$ and $\mathbb{Z}_2^T$ anti-commute when they act on the end points.

  6. $\mathbb{Z}_2^{(y)}$ and $\mathbb{Z}_2^T$ anti-commute when they act on the end points.

With this understanding, we can fully pin down the anomaly of the Kitaev spin-$S$ model. Clearly, there cannot be any anomaly solely associated with the ordinary 0-form symmetries, because they are on-site and necessarily anomaly-free. On the other hand, by checking the end point statistics, we have found a nontrivial (trivial) anomaly solely associated with the $\mathbb{Z}_2^{[1]}$ symmetry if $S \in \mathbb{Z} + \frac{1}{2}$ ($S \in \mathbb{Z}$). All we need to do is to understand whether there is any mixed anomaly between the 0-form $\mathbb{Z}_2^{(x)} \times \mathbb{Z}_2^{(y)} \times \mathbb{Z}_2^T$ symmetry and the 1-form $\mathbb{Z}_2^{[1]}$ symmetry. As argued above, this mixed anomaly is nontrivial if the $\mathbb{Z}_2^{(x)} \times \mathbb{Z}_2^{(y)} \times \mathbb{Z}_2^T$ symmetry is fractionalized at the end points of the strings associated with the $\mathbb{Z}_2^{[1]}$ symmetry.

## 4.1 The case with $S = 1/2$

Below we show that the $\mathbb{Z}_2^{(x)} \times \mathbb{Z}_2^{(y)} \times \mathbb{Z}_2^T$ is not fractionalized for $S = 1/2$, and later we will see that this is enough to determine the mixed anomaly for all $S$. When $S = 1/2$, the Kitaev model can be exactly solved [1]. The solution is based on a parton construction, where at each site one introduces 4 species of Majorana fermions, $\gamma^{0,x,y,z}$, such that the spin operator can be written as $S^\mu = i\gamma^\mu \gamma^0$. In this case, the strings associated with the $\mathbb{Z}_2^{[1]}$ symmetry are precisely the Wilson lines of these Majorana fermions, so these Majorana fermions should be identified as the end points of the strings. Then we only need to examine whether the $\mathbb{Z}_2^{(x)} \times \mathbb{Z}_2^{(y)} \times \mathbb{Z}_2^T$ symmetry is fractionalized on these Majorana fermions. The action of the $\mathbb{Z}_2^{(x)} \times \mathbb{Z}_2^{(y)} \times \mathbb{Z}_2^T$ symmetry on the Majorana fermions are given by Table 1 [45]. By comparing Table 1 against the 6 distinct fractionalization patterns, we see that the $\mathbb{Z}_2^{(x)} \times \mathbb{Z}_2^{(y)} \times \mathbb{Z}_2^T$ is not fractionalized. So there is no mixed anomaly between the 0-form $\mathbb{Z}_2^{(x)} \times \mathbb{Z}_2^{(y)} \times \mathbb{Z}_2^T$ symmetry and 1-form $\mathbb{Z}_2^{[1]}$ symmetry.

Therefore, the Kitaev spin-1/2 model has an anomaly purely associated with the $\mathbb{Z}_2^{[1]}$ 1-form symmetry, but there is no anomaly associated with the 0-form symmetry or mixed anomaly between the 0-form and 1-form symmetries.

## 4.2 Even-odd effect

Next, we discuss the anomaly for general $S$. First, because the anomalies are classified by $\mathbb{Z}_2^{17}$, an even number of copies of Kitaev spin-1/2 model is non-anomalous. Below, we will

Table 1: Action of the $\mathbb{Z}_2^{(x)} \times \mathbb{Z}_2^{(y)} \times \mathbb{Z}_2^T$ symmetry on the Majorana fermions in the Kitaev spin-1/2 model. Here $A$ and $B$ label the sublattices (see Fig. 1 (a)).

|  | $\mathbb{Z}_2^{(x)}$ | $\mathbb{Z}_2^{(y)}$ | $\mathbb{Z}_2^T$ |
|---|---|---|---|
| $\gamma_A^x \rightarrow$ | $\gamma_A^x$ | $-\gamma_A^x$ | $\gamma_A^x$ |
| $\gamma_B^x \rightarrow$ | $\gamma_B^x$ | $-\gamma_B^x$ | $-\gamma_B^x$ |
| $\gamma_A^y \rightarrow$ | $-\gamma_A^y$ | $\gamma_A^y$ | $\gamma_A^y$ |
| $\gamma_B^y \rightarrow$ | $-\gamma_B^y$ | $\gamma_B^y$ | $-\gamma_B^y$ |
| $\gamma_A^z \rightarrow$ | $-\gamma_A^z$ | $-\gamma_A^z$ | $\gamma_A^z$ |
| $\gamma_B^z \rightarrow$ | $-\gamma_B^z$ | $-\gamma_B^z$ | $-\gamma_B^z$ |
| $\gamma_A^0 \rightarrow$ | $\gamma_A^0$ | $\gamma_A^0$ | $\gamma_A^0$ |
| $\gamma_B^0 \rightarrow$ | $\gamma_B^0$ | $\gamma_B^0$ | $-\gamma_B^0$ |

construct an interpolation between $2S$ decoupled copies of the Kitaev spin-1/2 model and a Kitaev spin-$S$ model, such that all symmetries of the Kitaev spin-$S$ model are preserved along the entire interpolation. This means that the anomaly of the Kitaev spin-$S$ model is equivalent to the anomaly of $2S$ copies of the Kitaev spin-1/2 model. Therefore, we get an even-odd effect: Models for all $S \in \mathbb{Z} + \frac{1}{2}$ have the same anomaly as the Kitaev spin-1/2 model, and models for all $S \in \mathbb{Z}$ are non-anomalous.

To construct this interpolation, consider a honeycomb lattice system, where at each site there are $2S$ species of spin-1/2 moments. The interpolation of the Hamiltonians is given by

$$H(\xi) = (1 - \xi)H_1 + \xi H_2, \quad \xi \in [0, 1], \tag{14}$$

with

$$
\begin{aligned}
H_1 &= -\sum_{\alpha=1}^{2S} \sum_{\langle i,j \rangle \in \mu} J_1^\mu s_{\alpha,i}^\mu s_{\alpha,j}^\mu, \\
H_2 &= -J_2 \sum_i \left( \sum_{\alpha=1}^{2S} \boldsymbol{s}_{\alpha i} \right) \left( \sum_{\beta=1}^{2S} \boldsymbol{s}_{\beta i} \right) - \sum_{\langle i,j \rangle \in \mu} J_3^\mu \left( \sum_{\alpha=1}^{2S} s_{\alpha i}^\mu \right) \left( \sum_{\beta=1}^{2S} s_{\beta j}^\mu \right),
\end{aligned}
\tag{15}
$$

where $s_{\alpha i}^\mu$ is a spin-1/2 operator at site $i$ for species $\alpha$, and $J_2 \gg |J_3^{x,y,z}|$. It is straightforward to check that i) $H(0)$ is the Hamiltonian of $2S$ decoupled copies of the Kitaev spin-1/2 model, ii) $H(1)$ is effectively the Hamiltonian of the Kitaev spin-$S$ model, Eq. (1), where the spin operator in Eq. (1) is identified as $S_i^\mu = \sum_{\alpha=1}^{2S} s_{\alpha i}^\mu$,[2] and iii) all symmetries of the Kitaev spin-$S$ model are preserved for any $\xi \in [0, 1]$. Therefore, this interpolation fulfills our purpose.

## 5 Consequences of the symmetries and anomalies

Having determined the anomaly associated with the $\mathbb{Z}_2^{[1]} \times \mathbb{Z}_2^{(x)} \times \mathbb{Z}_2^{(y)} \times \mathbb{Z}_2^T$ symmetry in the Kitaev spin-$S$ models, in this section we discuss the consequences of the symmetry and anomaly. These consequences apply to not only the Kitaev spin-$S$ model in Eq. (1), but also any of its perturbed versions, as long as the perturbations are local and preserve the relevant symmetry.

---

[2] Because $J_2 \gg |J_3^{x,y,z}|$, we can first ignore $J_3^{x,y,z}$ and consider only the $J_2$ term. The $J_2$ term forces all $2S$ species of spin-1/2's to form a spin-$S$ moment. Within the low-energy Hilbert space made of these spin-$S$ moments, the $J_3^{x,y,z}$ term precisely gives the Kitaev model.

An example of such perturbations is the single-ion anisotropy, with $\delta H = D \sum_i (S_i^z)^2$. In this context, the consequences we discuss below apply to arbitrarily large $D$. In fact, in Sec. 5.3 we will further argue that these consequences are robust even if the local perturbations break the $\mathbb{Z}_2^{[1]}$ symmetry, as long as these perturbations are weak.

## 5.1   $S \in \mathbb{Z} + \frac{1}{2}$

Let us start with the case where $S \in \mathbb{Z} + \frac{1}{2}$. First, due to the nontrivial anomaly, the ground state cannot be short-range entangled. Moreover, the anomaly of the $\mathbb{Z}_2^{[1]}$ symmetry implies that at low energies there are generically deconfined fermionic excitations, such that the bound state of two such fermions is an ordinary local excitation. These fermions can be created by applying open string operators to the ground state. In the field theoretic language, this means that the low-energy effective field theory contains a one dimensional topological defect with topological spin $-1$. In addition, due to the absence of mixed anomaly between the 1-form and 0-form symmetries, this fermion carries no fractional quantum number under the 0-form symmetry.

Put in short, the symmetries and anomalies in this case imply that the system realizes symmetry-enforced exotic quantum matter.

Examples of quantum phases satisfying the above constraints are familiar in the Kitaev spin-1/2 model, which include a gapless phase described by Majorana fermions coupled to a dynamical $\mathbb{Z}_2$ gauge field, and a $\mathbb{Z}_2$ topological order. If the $\mathbb{Z}_2^T$ symmetry is broken, a non-Abelian Ising topological order can also emerge, where the fermionic excitation carries no fractional quantum number under the $\mathbb{Z}_2^{(x)} \times \mathbb{Z}_2^{(y)}$ symmetry [1]. An example of perturbation that breaks $\mathbb{Z}_2^T$ but preserves $\mathbb{Z}_2^{(x)} \times \mathbb{Z}_2^{(y)} \times \mathbb{Z}_2^{[1]}$ is the 3-spin interaction in Ref. [1] (also see Appendix B). In all these quantum phases, the low-energy effective field theory contains an anomalous $\mathbb{Z}_2$ 1-form symmetry, coming from the microscopic $\mathbb{Z}_2^{[1]}$ symmetry. Since for all $S \in \mathbb{Z} + \frac{1}{2}$, the models have the same anomaly, the hypothesis of emergibility [46,47] suggests that all these examples of quantum phases can emerge either in the Kitaev spin-$S$ model, or by perturbing it in a symmetry-preserving manner.

Besides the above quantum phases that are known to arise in the Kitaev spin-1/2 model, there can be additional ones which can be obtained by appropriate symmetric perturbations to the model. For example, one interesting quantum phase is where the ordinary $\mathbb{Z}_2^{(x)} \times \mathbb{Z}_2^{(y)} \times \mathbb{Z}_2^T$ symmetry is spontaneously broken, so that the corresponding model realizes coexistence of deconfined fractional fermionic excitations and conventional spontaneous symmetry breaking. It is also possible to realize a low-energy theory where the deconfined fermionic excitations undergo a Gross-Neveu-Yukawa type quantum phase transition, while being coupled to a dynamical $\mathbb{Z}_2$ gauge field. This quantum phase transition may connect a quantum phase where the ordinary $\mathbb{Z}_2^{(x)} \times \mathbb{Z}_2^{(y)} \times \mathbb{Z}_2^T$ symmetry is spontaneously broken and another quantum phase where it is not. To pin down which microscopic models give rise to these quantum phases and phase transitions requires extensive numerical studies, and it is beyond the scope of this paper.

## 5.2   $S \in \mathbb{Z}$

Next, we turn to the case where $S \in \mathbb{Z}$. To simplify the discussion and to be physically relevant, we will focus on the case without fine tuning, which rules out scenarios like those, for example, discussed in Ref. [48]. One of the implications of the absence of fine tuning is that the symmetry generated by each individual $W_p$ operator (such as Eq. (2)) is not spontaneously broken.[3] Namely, no matter whether the system is defined on an infinite disk or torus, the

---

[3]This type of spontaneous symmetry breaking requires fine tuning because a symmetric local perturbation proportional to $W_p$ can lift the ground state degeneracy.

ground state is an eigenstate of the $W_p$ operator defined for each plaquette. However, when the system is defined on a torus, we do not assume that the ground state must be unique, and it is possible that the symmetry generated by the operators in Eq. (3) is spontaneously broken.

In this case, first of all, the absence of any anomaly implies that a $\mathbb{Z}_2^{[1]} \times \mathbb{Z}_2^{(x)} \times \mathbb{Z}_2^{(y)} \times \mathbb{Z}_2^T$ symmetric short-range entangled ground state is possible. But how is such a ground state compatible with the fact that applying open string operators to it creates a pair of excitations that seem to be fractional and deconfined, because these strings are tensionless? The resolution is that the end points of these strings are bosons, as required by the (absence of) $\mathbb{Z}_2^{[1]}$ anomaly, and in such a symmetric short-range entangled ground state these bosons are condensed and give rise to no deconfined fractional excitation.[4]

There are further consequences if it is known that the $\mathbb{Z}_2^{[1]}$ symmetry is spontaneously broken, which means there are multiple degenerate ground states if the system is defined on a torus, such that these ground states are eigenstates of the operators in Eq. (3) with different eigenvalues. In this case, there will be two types of deconfined excitations. One of them are bosonic excitations that can be created by applying to the ground states the open string operators associated with the $\mathbb{Z}_2^{[1]}$ symmetry. These bosonic excitations are not themselves ordinary local excitations, but the bound state of a pair of them is. Moreover, these bosonic excitations should carry no fractional quantum number under the 0-form $\mathbb{Z}_2^{(x)} \times \mathbb{Z}_2^{(y)} \times \mathbb{Z}_2^T$ symmetry, because of the absence of mixed anomaly between the $\mathbb{Z}_2^{[1]}$ and $\mathbb{Z}_2^{(x)} \times \mathbb{Z}_2^{(y)} \times \mathbb{Z}_2^T$ symmetries. Why are these bosons not condensed in this case? This is because the spontaneous breaking of the $\mathbb{Z}_2^{[1]}$ symmetry implies the presence of the other deconfined excitation, which is gapped and has $\pi$ mutual braiding statistics with these bosonic excitations [30–32]. In the lattice, states with this other type of excitations are eigenstates of some $W_p$ operators that have different eigenvalues as the ground state. An example of such a quantum phase is a $\mathbb{Z}_2$ topological order.

## 5.3  Robustness of the consequences

Because the above reasoning is purely based on symmetries and anomalies, the consequences we obtain are clearly generally applicable even if we perturb the Kitaev spin-$S$ model by local perturbations that respect the $\mathbb{Z}_2^{[1]} \times \mathbb{Z}_2^{(x)} \times \mathbb{Z}_2^{(y)} \times \mathbb{Z}_2^T$ symmetry. We remark that these consequences still generically apply even if the $\mathbb{Z}_2^{[1]}$ symmetry is weakly broken by local perturbations [31, 49–53]. A simple way to see it is to consider the effective field theory for the underlying quantum phase. In the effective field theory, all operators transforming nontrivially under the 1-form symmetry are supported on 1-dimensional manifolds, i.e., they are not local operators in the field theory [30]. So the perturbed theory still has an emergent 1-form symmetry at low energies since no local perturbation can break it, and all the aforementioned constraints still apply. Only when the perturbation is strong enough so that the original effective field theory fails to describe the lattice system, these constraints will cease to apply.

The above consideration holds when the excited states that have different $W_p$ eigenvalues compared to the ground states have a finite energy gap. The critical local perturbation that makes the original effective field theory fail is of the order of this gap. If this gap happens to be vanishing in the absence of perturbation, the consequences of the $\mathbb{Z}_2^{[1]}$ symmetry are not necessarily stable against an infinitesimal local perturbation, and the existence of deconfined fermionic excitations should also be more carefully justified. However, generically this gap is finite unless the Hamiltonian is fine tuned, so all our conclusions are valid for almost all local Hamiltonians with the symmetries and anomalies discussed above.

---

[4]Here an excitation is condensed if the long open string operators creating them have nonzero expectation values in the ground states.

To make this discussion less abstract, let us consider a concrete and familiar example, i.e., the Kitaev spin-1/2 model in the isotropic limit, where $J_x = J_y = J_z$. For simplicity, we will ignore the $\mathbb{Z}_2^{(x)} \times \mathbb{Z}_2^{(y)} \times \mathbb{Z}_2^T$ symmetry, and focus on the consequence due to the $\mathbb{Z}_2^{[1]}$ symmetry, i.e., the presence of deconfined fermionic excitations. This model has an exact $\mathbb{Z}_2^{[1]}$ symmetry. The low-energy effective theory is described by Majorana fermions coupled to a gapped dynamical $\mathbb{Z}_2$ gauge field, where the fermions are gapless [1]. In this field theory, the presence of a $\mathbb{Z}_2$ 1-form symmetry can be attributed to the gap of flux excitations of the dynamical $\mathbb{Z}_2$ gauge field.

Now suppose we perturb the model by a weak magnetic field, then the same effective field theory still applies, despite it is in a different regime where the Majorana fermions are gapped [1]. In this case, although the lattice system no longer has an exact $\mathbb{Z}_2^{[1]}$ symmetry, the effective field theory still has an emergent $\mathbb{Z}_2$ 1-form symmetry because the gap of the $\mathbb{Z}_2$ gauge flux cannot close by an infinitesimal perturbation, and the consequence of the $\mathbb{Z}_2^{[1]}$ symmetry, i.e., the presence of the deconfined fermionic excitations, still applies. When this perturbation theory is strong enough to close the gap of the $\mathbb{Z}_2$ gauge flux, the effective field theory is no longer described by Majorana fermions coupled to a $\mathbb{Z}_2$ gauge field, and in this case the consequences discussed above do not apply any more. In fact, if the magnetic field is very strong, the ground state is simply a fully polarized state, which indeed hosts no deconfined fermionic excitations.

# 6 Discussion

In this paper, we have analyzed the symmetries and anomalies of the Kitaev spin-$S$ models, and discussed their profound physical consequences. In short, the Kitaev spin-$S$ models have a $\mathbb{Z}_2^{[1]} \times \mathbb{Z}_2^{(x)} \times \mathbb{Z}_2^{(y)} \times \mathbb{Z}_2^T$ symmetry, and a nontrivial anomaly occurs only if $S \in \mathbb{Z} + \frac{1}{2}$, which is purely associated with the $\mathbb{Z}_2^{[1]}$ symmetry. The symmetry and anomaly have various consequences in the ground state and the low-energy excitations, as discussed in Sec. 5. In particular, if $S \in \mathbb{Z} + \frac{1}{2}$ the system realizes symmetry-enforced exotic quantum matter with emergent fermions.

On one hand, our results can be viewed as constraints on the ground states of the Kitaev spin-$S$ models from their symmetries and anomalies. To fully understand their ground states, however, one has to go beyond our analysis, and most likely numerical studies are needed. On the other hand, if one is interested in the phase diagram containing all models with these symmetries and anomalies, which include the Kitaev model as a special example, our results provide the basis to classify all quantum phases that can emerge on this phase diagram. The hypothesis of emergibility in Ref. [46] conjectures that all quantum phases that can match the anomaly can emerge somewhere on the phase diagram, and based on this idea classifications of various exotic quantum phases have been performed [47, 54]. It is of interest to apply this idea to the Kitaev models and their perturbed versions.

Also, in this paper we focus on the internal symmetries, and it is interesting to incorporate lattice symmetries into the analysis. Furthermore, when determining the mixed anomaly between the 0-form and 1-form symmetries, we referred to the solvable Kitaev model. It is useful to develop a method that can determine the anomaly without using any Hamiltonian. In addition, it is important to identify numerical and experimental probes for the emergent fractional excitations in these systems. Finally, since our idea of inferring the existence of fractional excitations from symmetries and anomalies is quite general, applying it to other setups may also lead to profound insights. We leave these to future work.

We also remark that the present work is not only of conceptual importance, but also of practical interest. One lesson from this work (and also Refs. [33–35]) is that certain symmetries of a system can enforce this system to realize an exotic quantum phase of matter, which features, for example, fractional excitations. Therefore, to experimentally realize such exotic quantum phases of matter, it is helpful to first identify experimental setups where such symmetries are present, at least approximately.

## Acknowledgments

We thank Hank Chen, Max Metlitski, Sahand Seifnashri, Pok Man Tam and Chong Wang for useful discussions.

**Funding information** HTL was supported in part by the Packard Foundation and the Center for Theoretical Physics at MIT. He would like to thank the Perimeter Institute for Theoretical Physics for their hospitality during the course of this research. Research at Perimeter Institute is supported in part by the Government of Canada through the Department of Innovation, Science and Industry Canada and by the Province of Ontario through the Ministry of Colleges and Universities.

## A  General discussion of the lattice version of a $\mathbb{Z}_n$ 1-form symmetry in two spatial dimensions

In this appendix, we will discuss some general aspects of the lattice version of a $\mathbb{Z}_n$ 1-form symmetry. Specifically, in Appendix A.1 we discuss the important algebraic relations the operators defining a 1-form symmetry must satisfy, in Appendix A.2 we elaborate on the statistical phase factor of the end points of strings associated with a 1-form symmetry, in Appendix A.3 we show that such a nontrivial statistical phase factor implies an obstruction to gauging the 1-form symmetry, and in Appendix A.4 we illustrate a $\mathbb{Z}_n$ 1-form symmetry in a model discussed in Ref. [55]. To be concrete and to be related to the Kitaev models, we focus on a honeycomb lattice where at each of its vertex there is a localized spin-$S$ degree of freedom. But our discussions can be generalized to other types of lattice systems.

### A.1  General algebraic relations

In the field theoretic context, the generators of a 1-form symmetry in $2+1$ spacetime dimensions should be supported on closed loops. These generators should commute with each other, and the closed loops supporting them can be arbitrarily deformed. As discussed in the main text, the lattice versions of these two conditions are satisfied by the $\mathbb{Z}_2^{[1]}$ 1-form symmetry of the Kitaev spin-$S$ model. Below, we discuss the constraints these two conditions impose in the context of a general $\mathbb{Z}_n$ 1-form symmetry on a honeycomb lattice spin-$S$ system.

Suppose the generator of the $\mathbb{Z}_n$ 1-form symmetry supported on the plaquette labeled by $p$ in Fig. 4 is

$$W_p = \eta_p \, O_{p,1}^{(1)} O_{p,2}^{(2)} O_{p,3}^{(3)} O_{p,4}^{(4)} O_{p,5}^{(5)} O_{p,6}^{(6)}. \tag{A.1}$$

In the above, $O_{p,i}^{(j)}$ is a unitary operator supported at the $i$-th vertex of the plaquette $p$, and the superscript $j$ labels a specific unitary operator at this vertex. For example, $O_{p,1}^{(3)}$ denotes the operator obtained by translating $O_{p,3}^{(3)}$ from vertex 3 to vertex 1. Also note that in this notation,

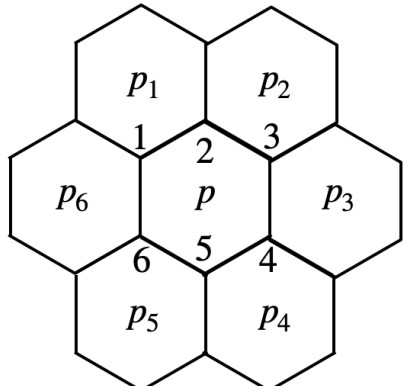

Figure 4: A plaquette labeled by $p$ is surrounded by 6 other plaquettes, labeled by $p_1$, $p_2$, $\cdots$, and $p_6$, respectively. The 6 vertices of the plaquette $p$ are labeled by numbers 1, 2, $\cdots$, and 6, respectively. The vertices of other plaquettes are ordered and indexed in the same way as those of plaquette $p$.

a single vertex may have multiple labels. For example, "$p, 1$", "$p_1, 5$" and "$p_6, 3$" label the same vertex. The $U(1)$ phase factor $\eta_p$ in the definition of $W_p$ is the eigenvalue of the ground states with respect to $(W_p/\eta_p)^\dagger$.[5] It is introduced in the definition so that $W_p$ acts as the identity operator on the ground states. The $\mathbb{Z}_n$ nature of the 1-form symmetry demands that $(W_p)^n$ be a $U(1)$ phase factor, which then implies that $\left(O^{(j)}\right)^n = e^{i\phi_j}$. Without loss of generality, we can set $\left(O^{(j)}\right)^n = 1$ by redefining $O^{(j)}$ into $e^{-i\phi_j/n}O^{(j)}$. This redefinition may change the value of $\eta_p$. But later we will see that all important aspects of the 1-form symmetry do not depend on the value of $\eta_p$. In fact, $\eta_p$ will not show up at all in the discussion below.

Now we discuss the constraints on these operators $O$'s. Because each plaquette is adjacent to 6 other plaquettes, and each pair of adjacent plaquettes share two common vertices, for $W_p$'s on different plaquettes to commute with each other, we demand

$$
\begin{aligned}
O_{p,1}^{(1)} O_{p,2}^{(2)} O_{p,2}^{(4)} O_{p,1}^{(5)} &= O_{p,2}^{(4)} O_{p,1}^{(5)} O_{p,1}^{(1)} O_{p,2}^{(2)}, \\
O_{p,2}^{(2)} O_{p,3}^{(3)} O_{p,3}^{(5)} O_{p,2}^{(6)} &= O_{p,3}^{(5)} O_{p,2}^{(6)} O_{p,2}^{(2)} O_{p,3}^{(3)}, \\
O_{p,3}^{(3)} O_{p,4}^{(4)} O_{p,4}^{(6)} O_{p,3}^{(1)} &= O_{p,4}^{(6)} O_{p,3}^{(1)} O_{p,3}^{(3)} O_{p,4}^{(4)}.
\end{aligned}
\tag{A.2}
$$

In the above, equations such as $O_{p,1}^{(j)} = O_{p_1,5}^{(j)}$ and $O_{p,2}^{(j)} = O_{p_2,6}^{(j)}$ have been used.

The above equations impose strong constraints on these operators. For example, the first equation can be written as

$$
O_{p,1}^{(1)} O_{p,1}^{(5)} O_{p,1}^{(1)\dagger} O_{p,1}^{(5)\dagger} = O_{p,2}^{(4)} O_{p,2}^{(2)} O_{p,2}^{(4)\dagger} O_{p,2}^{(2)\dagger}.
\tag{A.3}
$$

Because the unitary operators on the two sides of the above equation are supported at different vertices, both operators must be a $U(1)$ phase factor. We denote this $U(1)$ phase factor by $\alpha_1$. Now we can remove the subscripts indexing the plaquette and vertex, and we get the follow operator identities valid at each vertex of the lattice

$$
O^{(1)} O^{(5)} O^{(1)\dagger} O^{(5)\dagger} = O^{(4)} O^{(2)} O^{(4)\dagger} O^{(2)\dagger} = \alpha_1.
\tag{A.4}
$$

---

[5]There can be fine-tuned cases where the Hamiltonian has degenerate ground states that are eigenstates of $W_p$'s with different eigenvalues. Such cases are fine-tuned because local perturbations proportional to $W_p$'s can lift the degeneracy. In these fine-tuned cases, our conclusion applies to each ground state, which has its own set of $\eta_p$'s.

Similarly, we obtain

$$
\begin{aligned}
O^{(2)} O^{(6)} O^{(2)\dagger} O^{(6)\dagger} &= O^{(5)} O^{(3)} O^{(5)\dagger} O^{(3)\dagger} = \alpha_2 \,, \\
O^{(3)} O^{(1)} O^{(3)\dagger} O^{(1)\dagger} &= O^{(6)} O^{(4)} O^{(6)\dagger} O^{(4)\dagger} = \alpha_3 \,,
\end{aligned}
\tag{A.5}
$$

where $\alpha_{2,3}$ are also some $U(1)$ phase factors.

Next, we turn to the condition that the closed loops supporting these generators can be deformed. On a honeycomb lattice, this condition means that, when the three operators coming from three plaquettes that share a common vertex are multiplied, the product of the operators at this common vertex should be a $U(1)$ phase factor. Following similar reasoning as before, from this condition we get

$$
O^{(1)} O^{(3)} O^{(5)} = \beta_1 \,, \quad O^{(2)} O^{(4)} O^{(6)} = \beta_2 \,,
\tag{A.6}
$$

where $\beta_{1,2}$ are $U(1)$ phase factors. Because of Eqs. (A.4) and (A.5), the orders of the operators in Eq. (A.6) are unimportant, in the sense that changing their orders will just change $\beta_{1,2}$ into some other $U(1)$ phase factors.

Eq. (A.6) can be used to show that $\alpha_1 = \alpha_2 = \alpha_3$. To see it, note

$$
\begin{aligned}
\alpha_1 &= O^{(1)} O^{(5)} O^{(1)\dagger} O^{(5)\dagger} = \beta_1^* O^{(1)} O^{(5)} O^{(3)} = \beta_1^* \alpha_2 O^{(1)} O^{(3)} O^{(5)} = \alpha_2 \,, \\
\alpha_2 &= O^{(2)} O^{(6)} O^{(2)\dagger} O^{(6)\dagger} = \beta_2^* O^{(2)} O^{(6)} O^{(4)} = \beta_2^* \alpha_3 O^{(2)} O^{(4)} O^{(6)} = \alpha_3 \,.
\end{aligned}
\tag{A.7}
$$

For this reason, we denote $\alpha_1 = \alpha_2 = \alpha_3 = \alpha$. Later we will see that $\alpha$ is the statistics of the end points of the strings associated with this 1-form symmetry, and it characterizes the anomaly associated with the $\mathbb{Z}_n$ 1-form symmetry. Because $\left(O^{(j)}\right)^n = 1$ for any $j = 1, 2, \cdots, 6$, we have $O^{(5)} = \left(O^{(1)}\right)^n O^{(5)} = \alpha^n O^{(5)} \left(O^{(1)}\right)^n = \alpha^n O^{(5)}$. Therefore, in our setup $\alpha^n = 1$, i.e., $\alpha = e^{2\pi i \frac{m}{n}}$. Suppose the dimension of the Hilbert space at each site is $N$. By taking the determinants of both sides of Eq. (A.4), we get $\alpha^N = e^{2\pi i \frac{m}{n} N} = 1$. So for a given dimension of the local Hilbert space, the types of 1-form symmetry anomalies that can be realized are restricted, namely $m$ is an integer multiple of $\frac{n}{\gcd(n,N)}$.

Putting the above discussions together, the two conditions (i.e., (i) the generators of the $\mathbb{Z}_n$ 1-form symmetry commute with each other and (ii) the closed loops supporting these generators can be arbitrarily deformed) impose general algebraic relations among the operators $O$'s that define each generator as in Eq. (A.1), and these relations are given by Eqs. (A.4), (A.5) and (A.6). Moreover, the $U(1)$ phase factors in Eqs. (A.4) and (A.5) satisfy $\alpha_1 = \alpha_2 = \alpha_3 = \alpha$, with $\alpha^n = 1$. In addition, if the Hilbert space dimension at each site is $N$, then $\alpha^N = 1$. For the Kitaev spin-$S$ model, it is straightforward to verify that $\alpha = (-1)^{2S}$.

So far we have been focusing on the generators of the $\mathbb{Z}_n$ 1-form symmetry defined on each plaquette. If the system is defined on a torus, there are additional generators supported on the non-contractible loops of the torus, just like Eq. (3). These generators also lead to a constraint on the dimension of the local Hilbert space. The argument below is independent of the details of the lattice, unlike the constraint derived above, which holds only on a honeycomb lattice. For the $\mathbb{Z}_n$ 1-form symmetry discussed here, the generalizations of $W_1$ and $W_2$ in Eq. (3) are

$$
\begin{aligned}
W_1 &= O_7^{(2)} O_1^{(5)\dagger} O_2^{(2)} O_3^{(5)\dagger} O_8^{(2)} O_9^{(5)\dagger} \,, \\
W_2 &= O_{10}^{(4)\dagger} O_5^{(1)} O_4^{(4)\dagger} O_3^{(1)} O_8^{(4)\dagger} O_{11}^{(1)} \,.
\end{aligned}
\tag{A.8}
$$

It is straigthforward to check that $W_1$ and $W_2$ commute with $W_p$ for each plaquette, but they do not commute with each other. Instead, they satisfy $W_1 W_2 W_1^{-1} W_2^{-1} = \alpha^2$, which is a general consequence of the 1-form symmetry anomaly as $W_1 W_2 W_1^{-1} W_2^{-1}$ measures the full braiding of the 1-form symmetry end points. Suppose $\alpha = e^{2\pi i \frac{m}{n}}$ with $m \in \mathbb{Z}$, then the Hamiltonian on a

torus will have at least a degeneracy $\frac{n}{\gcd(n,2m)}$ for each energy level, where $\gcd(n,2m)$ denotes the greatest common divisor of $n$ and $2m$. So the dimension of the total Hilbert space must be an integer multiple of $\frac{n}{\gcd(n,2m)}$. On other other hand, if the dimension of the local Hilbert space at each site is $N$, the dimension of the total Hilbert space is $N^A$, where $A$ is the number of sites of the system. For these two conditions to be compatible, $N$ cannot be coprime with $\frac{n}{\gcd(n,2m)}$.

## A.2 Statistical phase factor of the end points

In the main text, using the approach depicted in Fig. 2, we have verified that the end points of the strings associated with the $\mathbb{Z}_2^{[1]}$ symmetry of the Kitaev spin-$S$ model have a self statistical phase factor $(-1)^{2S}$. Furthermore, we have verified that this phase factor remains the same if we change the locations of the end points or deform the shapes of the strings. In this subsection, using the algebraic relations among the operators defining a general $\mathbb{Z}_n$ 1-form symmetry discussed in Appendix A.1, we will show that i) The self statistics from this approach always gives a $U(1)$ phase factor, rather than a more nontrivial unitary operation, and ii) This $U(1)$ phase factor is unambiguous, i.e., it remains the same if we deform the shapes of the strings.[6] In fact, this statistical phase factor is precisely $\alpha$. These results imply that the self statistics determined from this approach is indeed well-defined for a general $\mathbb{Z}_n$ 1-form symmetry satisfying the conditions in Appendix A.1. Again, we remark that although this approach was originally designed for topological orders, in our context we do not need to make any assumption about the quantum phase realized by our Hamiltonian. For the Kitaev spin-$S$ model, $\alpha = (-1)^{2S}$, so the corresponding end points have fermionic (bosonic) statistics if $S \in \mathbb{Z} + \frac{1}{2}$ ($S \in \mathbb{Z}$).

As in the main text, let us first write down the operators $M_{1,2,3}$. In general, they take the form

$$
\begin{aligned}
M_1 &= U_4 O_3^{(1)} O_2^{(4)\dagger} O_1^{(1)} U_0^{(1)}, \\
M_2 &= U_0^{(2)} O_5^{(3)\dagger} O_6^{(6)} O_7^{(3)\dagger} U_8, \\
M_3 &= U_{12} O_{11}^{(5)} O_{10}^{(2)\dagger} O_9^{(5)} U_0^{(3)},
\end{aligned}
\tag{A.9}
$$

where $U_0^{(1,2,3)}$ are unitary operators supported at the vertex 0, and $U_4$, $U_8$ and $U_{12}$ are unitary operators supported at the vertex 4, 8 and 12, respectively. In writing down the above operators, Eq. (A.6) has been used.

We would like to evaluate $M_1 M_2 M_3 (M_3 M_2 M_1)^\dagger = U_0^{(1)} U_0^{(2)} U_0^{(3)} (U_0^{(3)} U_0^{(2)} U_0^{(1)})^\dagger$. Again, in order for the strings to seamlessly connect to become longer strings, we demand

$$
U_0^{(1)} U_0^{(2)} = \lambda_1 O_0^{(6)}, \quad U_0^{(3)} U_0^{(2)} = \lambda_2 O_0^{(2)\dagger},
\tag{A.10}
$$

where $\lambda_{1,2}$ are some $U(1)$ phase factors. In writing down the second equation, Eq. (A.6) has been used. Now we see that

$$
U_0^{(1)} U_0^{(2)} U_0^{(3)} (U_0^{(3)} U_0^{(2)} U_0^{(1)})^\dagger = \lambda_1 \lambda_2^* O_0^{(6)} U_0^{(3)} U_0^{(1)\dagger} O_0^{(2)} = O_0^{(6)} O_0^{(2)\dagger} O_0^{(6)\dagger} O_0^{(2)} = \alpha.
\tag{A.11}
$$

It is straightforward to check if the shapes of the strings are deformed in a way that preserves their relative positions as in Fig. 2 (a), the statistical phase factor we obtain is always $\alpha$, which shows that self statistics via this approach is indeed well-defined, for any $\mathbb{Z}_n$ 1-form symmetry obeying the conditions in Appendix A.1. By deforming the shapes of the strings, here we include the deformations that retain the operators in the interiors of the strings, but change

---

[6]It obviously remains the same if we simply change the locations of the end points.

the operators at their ends, such as the operators $U_0^{(1,2,3)}$, into operators that are supported on multiple sites in disk-like regions, which have linear sizes much smaller than the lengths of the strings themselves.

### A.3 Gauging the $\mathbb{Z}_n$ 1-form symmetry

In this subsection, we discuss how to gauge the $\mathbb{Z}_n$ 1-form symmetry. We will find an obstruction to this gauging procedure when $\alpha \neq 1$, which confirms that the nontrivial self statistics yields the anomaly of the $\mathbb{Z}_n$ 1-form symmetry.

It is straightforward to extend the gauging procedure in Sec. 3.2 to a general $\mathbb{Z}_n$ 1-form symmetry. The key in the gauging procedure is to identify the Gauss law. For a general $\mathbb{Z}_n$ 1-form symmetry, the analog of Eq. (11) is

$$
\begin{aligned}
G_k^{(1)} &= O_k^{(3)} E_k^x E_k^z, \\
G_k^{(2)} &= O_k^{(1)} E_k^y E_k^z, \\
G_k^{(3)} &= G_k^{(1)} G_k^{(2)},
\end{aligned}
\tag{A.12}
$$

where now the operators $E$ are generalized Pauli matrices for an $n$ dimensional Hilbert space defined on a link, which represents the $\mathbb{Z}_n$ gauge field. Again, because $O_k^{(3)}$ and $O_k^{(1)}$ do not commute unless $\alpha = 1$, there is an obstruction to gauging this $\mathbb{Z}_n$ 1-form symmetry unless $\alpha = 1$.

### A.4 The $\mathbb{Z}_n$ 1-form symmetry in the generalized Kitaev model

In this subsection, we illustrate a $\mathbb{Z}_n$ 1-form symmetry in the generalized Kitaev model proposed in Ref. [55]. This 1-form symmetry and its anomaly were not discussed in Ref. [55], but were discussed in Ref. [34]. Here we discuss them in the framework introduced above.

This model is defined on a honeycomb lattice spin-$S$ system with $n = 2S+1$. At each site, we can define a set of basis states of the local Hilbert space, denoted by $|j\rangle$, where $j = 1, 2, \cdots, n$. We further define operators $T^x$, $T^y$ and $T^z$ such that $T^x|j\rangle = |j+1 \pmod{n}\rangle$, $T^z|j\rangle = e^{\frac{2\pi i}{n} j}|j\rangle$, and $T^y = -iT^{z\dagger}T^{x\dagger}$.

In the definition of the $\mathbb{Z}_n$ 1-form symmetry generator, Eq. (A.1), we take $O^{(1)} = O^{(4)} = T^y$, $O^{(2)} = O^{(5)} = T^z$ and $O^{(3)} = O^{(6)} = T^x$. It is straightforward to check that $(W_p/\eta_p)^n = 1$, and the conditions Eqs. (A.4), (A.5) and (A.6) are all satisfied, with $\alpha = e^{\frac{2\pi i}{n}}$. According to the discussion above, this $\mathbb{Z}_n$ 1-form symmetry is anomalous, and the end points of the strings associated with this 1-form symmetry have self statistical phase factor $e^{\frac{2\pi i}{n}}$.

The Hamiltonian with this $\mathbb{Z}_n$ 1-form symmetry can be taken as

$$
H = \sum_{\langle i,j\rangle \in \mu} \left( J_\mu T_i^\mu T_j^\mu + \text{h.c.} \right),
\tag{A.13}
$$

where $\mu = x, y, z$ labels the three types of bonds, just like the standard Kitaev model, and $J_\mu$ is a bond-dependent parameter.

When $n = 2$, the model becomes the Kitaev spin-1/2 model, and the $\mathbb{Z}_n$ 1-form symmetry is precisely the $\mathbb{Z}_2^{[1]}$ symmetry in the present paper.

## B   Bordism classification of anomalies

In this appendix, we compute the classification of the anomalies associated with the internal $\mathbb{Z}_2^{(x)} \times \mathbb{Z}_2^{(y)} \times \mathbb{Z}_2^{[1]}$ symmetry and $\mathbb{Z}_2^T$ time reversal symmetry, where $G^{[p]}$ with group $G$ stands for a $p$-form symmetry. If a symmetry is 0-form, we do not explicitly write this superscript.

It is known that for a bosonic theory in $d$ spacetime dimensions, the relevant anomalies are classified by following (dual) bordism group[7] (see below for the definition) [56–58]

$$\hom(\Omega^O_{d+1}(B\mathbb{Z}_2^{(x)} \times B\mathbb{Z}_2^{(y)} \times B^2\mathbb{Z}_2); U(1)), \tag{B.1}$$

where $B^{p+1}G$ is the classifying space associated to $p$-form symmetry $G^{[p]}$, and the unorientedness is the *background field* for $\mathbb{Z}_2^T$. Given the spacetime manifold $M^d$, it can be shown that $(p+1)$-form background gauge fields for $G^{[p]}$ are in one-to-one correspondence to the homotopy class of maps $f : M^d \to B^{p+1}G$. These maps are called classifying maps.[8]

Let us recall the definition of bordism group $\Omega^O_{d+1}(X)$, where $X$ is a CW-complex. All manifolds will be unoriented in the following. An element in this group is a $(d+1)$-dimensional manifold $N^{d+1}$ equipped with a map $f : N^{d+1} \to X$ modding out the following equivalence relation: Given that $N^{d+1}$ and $N'^{d+1}$ are two manifolds defined above equipped with maps $f, f'$ respectively, if there is a $(d+2)$-dimensional manifold $W^{d+2}$ with a map $F : W^{d+2} \to X$ such that $\partial W^{d+2} = N^{d+1} \coprod N'^{d+1}$ and $F$ extends both $f, f'$, then we define $N^{d+1} \sim N'^{d+1}$. In short,

$$\Omega^O_{d+1}(X) := \frac{(d+1)\text{-manifolds with } f : N^{d+1} \to X}{\sim}. \tag{B.4}$$

The bordism class can be made into an abelian group where the group multiplication is given by disjoint union of manifolds. The group identity is the empty manifold.

If there are local fermions (in the UV), in order to classify the anomalies, one should replace unoriented manifolds with spin manifolds or $\text{Pin}^\pm$ manifolds.

It is also worth mentioning that for the unoriented case, all elements of $\Omega^O_*(X)$ are of order 2. This is because $N \times [0, 1]$ gives a bordism $N \coprod N \to \emptyset$.

Now we are ready to compute the bordism group (with $d = 3$ for Kitaev model). The main tool will be Atiyah-Hirzebruch spectral sequence (AHSS) [62]. The input (or $E^2$-page) of AHSS is given by

$$E^2_{p,q} := H_p(B\mathbb{Z}_2^{(x)} \times B\mathbb{Z}_2^{(y)} \times B^2\mathbb{Z}_2, \Omega^O_q(\mathrm{pt})), \tag{B.5}$$

where $\Omega^O_q(\mathrm{pt})$ is the unoriented bordism group of a point (see Ref. [63] for a detailed computation for it). We list the relevant ones below:

$$\Omega^O_q(\mathrm{pt}) = \begin{cases} \mathbb{Z}_2, & q = 0, \\ 0, & q = 1, \\ \mathbb{Z}_2, & q = 2, \\ 0, & q = 3, \\ \mathbb{Z}_2^2, & q = 4. \end{cases} \tag{B.6}$$

Substituting Eq. (B.6) into Eq. (B.5), we get the $E^2$-page as Fig. 5.

---

[7]For a group $G$, $\hom(G; U(1))$ is another group called the Pontryagin dual of $G$, denoted as $G^\vee$.

[8]Similar construction of the classifying spaces also applies to more general higher-group symmetries [59], where gauge transformation of background fields is modified by Green-Schwarz shift. There it also makes sense to talk about higher bundles and related classifying spaces [60,61]. For the special case of 2-group symmetry $\mathbb{G}$, we define $G = \pi_1(B\mathbb{G})$ and $A = \pi_2(B\mathbb{G})$, where $\pi_k$ denotes $k$-th homotopy group. In physical terms, $\pi_{k+1}(B\mathbb{G})$ is called $k$-form symmetry group. A 2-group symmetry is a mixing of 0-form symmetry and 1-form symmetry. Mathematically, we have a Postnikov decomposition

$$B^2 A \to B\mathbb{G} \to BG, \tag{B.2}$$

which is characterized by a Postnikov class $\beta \in H^3(BG, A)$. In our case, the 2-group splits, which means there is no mixing (i.e., Postnikov class $\beta = 0$). Hence

$$B\mathbb{G} \simeq B^2\mathbb{Z}_2 \times B\left(\mathbb{Z}_2^{(x)} \times \mathbb{Z}_2^y\right). \tag{B.3}$$

Note RHS is homotopically equivalent to $B^2\mathbb{Z}_2 \times B\mathbb{Z}_2^{(x)} \times B\mathbb{Z}_2^{(y)}$, as claimed earlier.

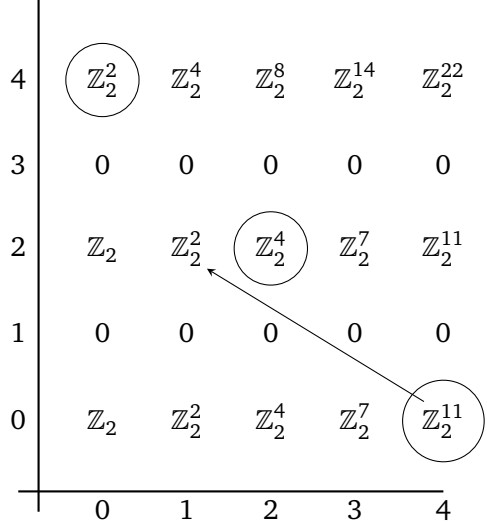

Figure 5: The $E^2$-page of AHSS in the unoriented case. The differential indicated in the figure vanishes.

In our case, since the Thom spectrum $MO$ is a graded Eilenberg-Maclane spectrum [64], the spectral sequence collapses at the $E^2$-page and there is no extension problem. As a result

$$\Omega_4^O(B\mathbb{Z}_2^{(x)} \times B\mathbb{Z}_2^{(y)} \times B^2\mathbb{Z}_2) = \bigoplus_{p+q=4} E_{p,q}^2 = \mathbb{Z}_2^{17}. \tag{B.7}$$

In summary, the anomalies associated with the $\mathbb{Z}_2^{(x)} \times \mathbb{Z}_2^{(y)} \times \mathbb{Z}_2^T \times \mathbb{Z}_2^{[1]}$ symmetry are classified by

$$\Omega_4^O(B\mathbb{Z}_2^{(x)} \times B\mathbb{Z}_2^{(y)} \times B^2\mathbb{Z}_2)^\vee = \mathbb{Z}_2^{17}. \tag{B.8}$$

Generators of the $\mathbb{Z}_2^{17}$ group are given by

$$\begin{aligned}
&w_1^4, \quad w_2^2, \quad a_1^4, \quad a_2^4, \quad w_1^2 a_1^2, \quad w_1^2 a_2^2, \quad a_1^3 a_2, \quad a_1^2 a_2^2, \quad a_1 a_2^3, \quad w_1^2 a_1 a_2, \\
&x_2^2, \quad a_1^2 x_2, \quad a_2^2 x_2, \quad a_1 a_2 x_2, \quad w_1^2 x_2, \quad a_1 w_1 x_2, \quad a_2 w_1 x_2.
\end{aligned} \tag{B.9}$$

In terms of the standard bulk-boundary correspondence for anomalies, the above 17 generators can be viewed as 17 topological actions of the 3+1 dimensional bulks, whose 2+1 dimensional boundaries have the 17 types of anomalies. Here $w_i$ is the $i$-th Stiefel-Whitney class of the tangent bundle of the spacetime manifold where the $3+1$ dimensional bulk lives on, $x_2$ is the background 2-form gauge field of $\mathbb{Z}_2^{[1]}$, and $a_1$ and $a_2$ are 1-form background gauge fields for $\mathbb{Z}_2^{(x)}$ and $\mathbb{Z}_2^{(y)}$, respectively. Notice that the first 10 of them are purely associated with the $\mathbb{Z}_2^{(x)} \times \mathbb{Z}_2^{(y)} \times \mathbb{Z}_2^T$ 0-form symmetry, the 11th of them is purely associated with the $\mathbb{Z}_2^{[1]}$ 1-form symmetry, and the last 6 of them are mixed anomalies between the 0-form and 1-form symmetries.

It is also of interest to obtain the classification of the anomalies associated with the $\mathbb{Z}_2^{(x)} \times \mathbb{Z}_2^{(y)} \times \mathbb{Z}_2^{[1]}$ symmetry. One way to get a model with such a symmetry is to add the following 3-spin interaction to the Kitaev spin-$S$ model:

$$H_{3\text{-spin}} = -h \sum_{j,k,l} S_j^x S_k^y S_l^z + (\text{symmetry related terms}), \tag{B.10}$$

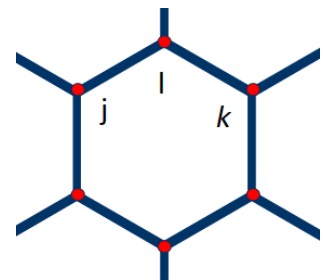

Figure 6: The 3-spin interaction Eq. (B.10) on sites $j$, $k$ and $l$.

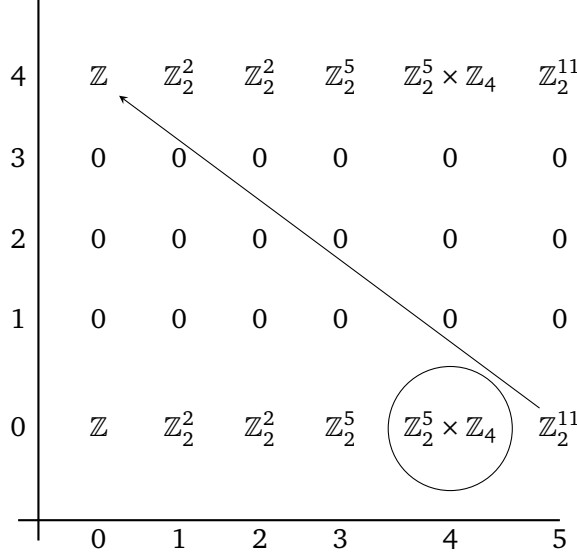

Figure 7: The $E^2$-page of AHSS without time reversal symmetry.

where $i, j, k$ are lattice sites arranged as in Fig. 6.

These anomalies are classified by the bordism group $\Omega_4^{SO}(B\mathbb{Z}_2^{(x)} \times B\mathbb{Z}_2^{(y)} \times B^2\mathbb{Z}_2)_{\text{tor}}$, where "tor" means taking the torsion part. The $E^2$-page in this case is given by Fig. 7.

For degree reasons, we can readily read the desired result in $E^2$-page:

$$\Omega_4^{SO}(B\mathbb{Z}_2^{(x)} \times B\mathbb{Z}_2^{(y)} \times B^2\mathbb{Z}_2)_{\text{tor}} = \mathbb{Z}_2^5 \times \mathbb{Z}_4 , \tag{B.11}$$

so

$$(\Omega_4^{SO}(B\mathbb{Z}_2^{(x)} \times B\mathbb{Z}_2^{(y)} \times B^2\mathbb{Z}_2)_{\text{tor}}^\vee = \mathbb{Z}_2^5 \times \mathbb{Z}_4 . \tag{B.12}$$

With the same notations in (B.9), generators of $\mathbb{Z}_2^5$ are respectively

$$a_1^2 x_2, a_2^2 x_2, a_1 a_2 x_2, a_1^3 a_2, a_1 a_2^3 , \tag{B.13}$$

and $\mathcal{P}(x_2)$ generates the $\mathbb{Z}_4$, where $\mathcal{P}$ is Pontryagin square.

Note that there is a natural map induced by inclusion $i : SO(n) \hookrightarrow O(n)$,

$$i^* : \Omega_{d+1}^O(X)^\vee \to \Omega_{d+1}^{SO}(X)^\vee , \tag{B.14}$$

for any $X$. Physically, this map tells us which $\mathbb{Z}_2^{(x)} \times \mathbb{Z}_2^{(y)} \times \mathbb{Z}_2^{[1]}$ anomaly a theory (e.g., the Kitaev spin-$S$ model perturbed by the 3-spin interaction in Eq. (B.10)) has, if this theory is

obtained by breaking the $\mathbb{Z}_2^T$ symmetry of another theory (e.g., the Kitaev spin-$S$ model) that has a $\mathbb{Z}_2^{(x)} \times \mathbb{Z}_2^{(y)} \times \mathbb{Z}_2^{[1]} \times \mathbb{Z}_2^T$ symmetry. In our situation,

$$i^*(x_2^2) = 2\mathcal{P}(x_2) \pmod 4, \tag{B.15}$$

and $i^*$ acts as the identity map on generators appearing in Eq. (B.13). All other terms that appear in Eq. (B.9) but not in Eq. (B.13) (such as $w_1^4$) vanish under $i^*$.

## C    Mixed anomalies between 0-form and 1-form symmetries from symmetry fractionalization

In this appendix, we explain the connection between symmetry fractionalization and the mixed anomaly between 0-form and 1-form symmetries. As discussed in Appendix B, these mixed anomalies are captured by the following topological actions of the $3+1$ dimensional bulk theories, whose boundaries realize the anomalies:

$$a_1^2 x_2, \; a_2^2 x_2, \; a_1 a_2 x_2, \; w_1^2 x_2, \; a_1 w_1 x_2, \; a_2 w_1 x_2. \tag{C.1}$$

We start with the first one, $a_1^2 x_2$. In the 3+1 dimensional spacetime manifold where the bulk theory lives, consider inserting a one-form symmetry defect on a two-dimensional submanifold $X$, ending on the boundary at a one dimensional submanifold $Y = \partial X$. It has the effect of turning on 2-form gauge field $x_2$ that is Poincaré dual to $X$, i.e., $x_2 = \delta(X_2)$. The topological action $a_1^2 x_2$ then contributes to the total action by $\int_X a_1^2 = \int_X \frac{1}{2} \delta a_1 = \int_Y \frac{1}{2} a_1$, where the Bockstein homomorphism is used. This action means that the boundary one-form symmetry line carries a half charge under the $\mathbb{Z}_2^{(x)}$ symmetry, i.e., this mixed anomaly is related to the fractionalization of the $\mathbb{Z}_2^{(x)}$ symmetry on the excitations living on the end points of the one-form symmetry line. Similar argument shows that the mixed anomaly $a_2^2 x_2$ is related to the fractionalization of the $\mathbb{Z}_2^{(y)}$ symmetry on the end points.

Next, we turn to $a_1 a_2 x_2$, using an argument similar to the one in Ref. [65]. Again, consider inserting a one-form symmetry defect of the $x_2$ gauge field on a two-dimensional submanifold $X$. The contribution from the topological action $a_1 a_2 x_2$ now becomes $\int_X a_1 a_2$, which means the surface $X$ in this case is decorated with a $1+1$ dimensional $\mathbb{Z}_2^{(x)} \times \mathbb{Z}_2^{(y)}$ symmetry-protected topological (SPT) state described by this topological action [40]. It is well known that the boundaries of this SPT carry a fractionalized projective representation of the $\mathbb{Z}_2^{(x)} \times \mathbb{Z}_2^{(y)}$ symmetry, such that $\mathbb{Z}_2^{(x)}$ and $\mathbb{Z}_2^{(y)}$ anti-commute when they act on the boundaries [66]. Therefore, the mixed anomaly described by $a_1 a_2 x_2$ implies this particular fractionalization pattern of the $\mathbb{Z}_2^{(x)} \times \mathbb{Z}_2^{(y)}$ symmetry on the end point excitations of the $\mathbb{Z}_2^{[1]}$ one-form symmetry line. Similar analysis shows that the anomalies described by the other 3 topological actions are related to the other fractionalization patterns of the $\mathbb{Z}_2^{(x)} \times \mathbb{Z}_2^{(y)} \times \mathbb{Z}_2^T$ symmetry on these end point excitations, as presented in the main text.

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
