# Peer review of "Symmetries and anomalies of Kitaev spin-S models: Identifying symmetry-enforced exotic quantum matter"

_SciPost Physics, doi:SciPost Phys. 16, 100 (2024)_

## Round 1 · Referee Report · Anonymous (Referee 1) · 2024-1-28

Strengths

1) The paper discusses the Kitaev spin-S model in the modern framework of generalized symmetries and, in doing so, provides a beautiful interpretation of the "even/odd" effect between $S\in\mathbb{Z}$ and $S\in\mathbb{Z}+\frac12$ in terms of 't Hooft anomalies. 2) The paper finds an interesting, yet simple, interpolation between 2S decoupled copies of the Kitaev spin-1/2 model and the Kitaev spin-S model. 3) The paper's appendices are nicely written, reviewing some relevant background material to the main text and elaborating on comments made in the main text, both with physical and mathematical purviews.

Weaknesses

1) The primary weakenss is that the paper minimally applies the symmetries and anomalies to explore new features of the model or related models. For example, section VI, which is supposed to discuss the physical consequences of the anomaly, is very brief. While it correctly states that the anomaly prevents trivial phases, it does not use this to learn anything new about the model.

Report

This paper explores the 't Hooft anomalies of ordinary and invertible finite 1-form symmetries in the Kitaev spin-S model. The authors diagnose the 't Hooft anomaly using a technique common in the study of topological order, showing that the anomaly is trivial for $S\in\mathbb{Z}$ while nontrivial for $S\in\mathbb{Z}+\frac12$. As they state in their paper, this implies that the Kitaev spin-S model, and any symmetric deformed versions, with $S\in\mathbb{Z}+\frac12$ cannot realize a trivial phase.

The paper is nicely written and has the benefit of discussing ideas common throughout the generalized symmetry literature in a way cond-mat physics folks outside this community should be able to follow and learn from. For this reason, the paper should ultimately be published. However, I would first like to see some minor revisions before publication. As I explain below, a few places in the main text contain minor errors (or perhaps need clarification). Furthermore, while I do not require related revisions, the paper lacks novelty at times, and I would love to see the authors expand on section VI (see suggestions below).

$\underline{\text{Some general comments/questions:}}$

(1) It would be helpful for the reader if the authors could summarize the results in Refs. 33-35 in more detail to contextualize their results with the established literature and emphasize the novelty of their work. I imagine one key result absent from the literature is the anomalies' dependency on S. If this is true, it would be helpful for the authors to state it clearly. One way the authors could improve the novelty of their paper is to include the lattice symmetries and study their interplay with the internal symmetries, which is currently stated as something left to future work. This interplay would be fascinating and be mathematically described by a nontrivial 2-group. This interplay would be related to what Kitaev called "weak symmetry breaking" in Appendix F of [1], and the group homomorphism $\omega_1: G \rightarrow \Gamma_1$ in Appendix F would be the defining data for the 2-group. Similar 2-group symmetries involving lattice symmetries were discussed in arXiv:2301.04706.

(2) It would be interesting if the authors could comment on modifications of the Kitaev spin-S model that would change the 't Hooft anomaly. For instance, modifications that would give rise to symmetry fractionalization patterns. For example, what I have in mind is something like appendix A of arXiv:2206.14197, which does this for the toric code and translation symmetry. It would be neat to see a similar exploration of the Kitaev spin-S model, which I believe lies in the scope of the paper and may be doable without importing fancy numerics techniques.

(3) It would be interesting if the authors could discuss the corresponding 3+1D SPT for their 't Hooft anomaly. I would imagine that the topological response theory is the well-known action $S = \mathrm{i}\pi\int \mathcal{A}\cup\mathcal{A}$, where $\mathcal{A}$ is a $\mathbb{Z}_2$ 2-cocycle. But it would be interesting to see this explored on the lattice by finding a 3+1D lattice Hamiltonian of the SPT whose boundary is the Kitaev spin-S model. Since the modern perspective of both the cond-mat and hep-th communities on 't Hooft anomalies of invertible symmetries is through its corresponding SPT, this feels like a relevant addition to the paper.

(4) As mentioned in the above weakness section, section VI is very brief. The authors correctly state the beautiful consequence of 't Hooft anomalies preventing the model from having a trivial phase, and they nicely contextualize this with known phases. I would love to see the authors explore interesting $\mathbb{Z}_2^{[1]}$ symmetric deformations of their model to investigate new interesting phases and further explore the hypothesis of emergibility. Another suggestion for the authors to expand section VI is to discuss how the microscopic anomalous $\mathbb{Z}_2^{[1]}$ symmetry is related to the IR symmetries in the different phases (particularly the $\mathbb{Z}_2$ and Ising topologically ordered phases). Some of these suggestions are mentioned in the paper's discussion section, but I feel it is a real shame not to explore some of them in the present paper.

$\underline{\text{Some specific comments/ questions:}}$

(1) In paragraph 2 of the introduction, the authors write, "The $\mathbb{Z}_2$ gauge field is deconfined if $S\in \mathbb{Z}+\frac12$, but it can be Higgsed if $S\in \mathbb{Z}$ [28]." I am confused about the wording here. Are they saying that for $S\in \mathbb{Z}+\frac12$, the model cannot undergo a Higgs transition into a trivial phase and is always in a deconfined phase regardless of perturbations added, while for $S\in \mathbb{Z}$ the model has both a deconfined and Higgs phase? I am also confused about what is being Higgsed here since the deconfined topological excitation discussed later on is a fermion. It would be helpful if the authors could reword this sentence to ensure their message is clearly communicated.

(2) It would be helpful for the reader if the authors could cite references [31, 46–48] after writing, "Moreover, even if the 1-form symmetry is slightly broken by local perturbations, its consequences are still robust" in the fifth paragraph of the introduction. Furthermore, there are a handful of other papers that should be cited alongside [31, 46–48] that make similar claims from different perspectives (e.g., arXiv:2106.12610, arXiv:2211.12543, arXiv:2202.05866, arXiv:2304.13751).

(3) When introducing Eq 2, The authors really should cite [1] since Kitaev wrote down this loop operator for the S=1/2 case (Eq. 6 of [1] with $\sigma^a = -\mathrm{i}\mathrm{e}^{\mathrm{i} \pi S^a}$) as well as [33] where the same operator $W_p$ is written down in the first paragraph of Sec. IIIC.

(4) The authors define 't Hooft anomaly in section III as any obstruction to gauging. They may want to be more specific in their definition since not all obstructions to gaugings are 't Hooft anomalies (e.g., one cannot gauge the $\mathbb{Z}_4/\mathbb{Z}_2$ part of a $\mathbb{Z}_4$ symmetry, which may seem silly to try but is a simplified example of obstructions to gauging parts of anomaly-free higher-group symmetries). In the context of invertible symmetries, 't Hooft anomalies are only obstructions to gauging when they can remedied by an SPT in one higher dimension.

(5) In the final paragraph of section III, the authors state that the anomaly can also be seen from the anisotropic limit. The calculation leading up to this paragraph is lovely and proves there is indeed an anomaly. I do agree that the fact that the anisotropic limit for $S\in \mathbb{Z}$ is trivial implies there is no anomaly, but I do not agree that just because it is nontrivial for $S\in \mathbb{Z}+\frac12$ that it means there is an anomaly. The latter observation is consistent with a 't Hooft anomaly and a sound check, but I don't think it proves there has to be one. Could the authors elaborate on this?

(6) In the second paragraph of section VI, the authors write, "...the anomaly of the $\mathbb{Z}^{[1]}_2$ symmetry implies that at low energies there are deconfined fermionic excitations..." Does this mean the authors claim that the 't Hooft anomaly can only be saturated by the $\mathbb{Z}^{[1]}_2$ symmetry spontaneously breaking? The claim is somewhat reasonable, but I don't see why it has to be true. Why is it not possible, for example, to add symmetric terms to the Kitaev spin-S model that can drive a phase transition to a gapless phase where the $\mathbb{Z}^{[1]}_2$ symmetry is not spontaneously broken?

(7) In the first paragraph of Appendix Section 3, the authors comment on the onsite aspect of the anomalous 1-form symmetry. I want to remark that the onsite versus non-onsite diagnosis for Hooft anomalies of generalized symmetries fails pretty often. For instance, In 1+1D Hamiltonian G gauge theory, where G is a finite group, the Rep(G) 0-form symmetry is anomaly-free but not always onsite (the symmetry operators are all matrix product operators). It is pretty standard that non-invertible symmetries of 1+1D Hamiltonian models are matrix product operators and are not on site (e.g., see arXiv:2112.09091)

(8) There is a typo in Eq. B1: $B \mathbb{Z}_2^{y)}$ should be $B \mathbb{Z}_2^{(y)}$. Also, it may be helpful for the mathematically inclined reader to explain that $B \mathbb{Z}_2^{(x)} \times B \mathbb{Z}_2^{(y)} \times B^2 \mathbb{Z}_2$ in Eq. B1 started its life as the classifying space of a 2-group $\mathbb{G}$ related to the split fibration $K(\mathbb{Z}_2,2)\to B\mathbb{G}\to K( \mathbb{Z}_2^{(x)} \times \mathbb{Z}_2^{(y)},1)$ in a Postnikov system.

Requested changes

There are what I believe to be a few minor mistakes pointed out in my report that I would like the authors to address (if they are indeed mistakes, that is!). Other than those, I do not have any requested changes, but I hope the authors give some thought to the other changes I suggested!

  • validity: good
  • significance: ok
  • originality: ok
  • clarity: high
  • formatting: good
  • grammar: excellent

Author:  Liujun Zou  on 2024-03-10  [id 4354]

(in reply to Report 1 on 2024-01-28)

We thank the referee for the comments and questions, which are helpful for us to improve our paper. These comments and suggestions are addressed below.

Report:

Strengths: 1) The paper discusses the Kitaev spin-S model in the modern framework of generalized symmetries and, in doing so, provides a beautiful interpretation of the "even/odd" effect between $S\in\mathbb{Z}$ and $S\in\mathbb{Z}+\frac{1}{2}$ in terms of 't Hooft anomalies. 2) The paper finds an interesting, yet simple, interpolation between 2S decoupled copies of the Kitaev spin-1/2 model and the Kitaev spin-S model. 3) The paper's appendices are nicely written, reviewing some relevant background material to the main text and elaborating on comments made in the main text, both with physical and mathematical purviews.

Response:

We thank the referee for pointing out these strengths of our work.

Report:

Weaknesses 1) The primary weakeness is that the paper minimally applies the symmetries and anomalies to explore new features of the model or related models. For example, section VI, which is supposed to discuss the physical consequences of the anomaly, is very brief. While it correctly states that the anomaly prevents trivial phases, it does not use this to learn anything new about the model.

Response:

As stated in the abstract and discussed in more detail in the paper, the consequence of the anomaly is not just the impossibility of a trivial phase, but also the existence of deconfined fermionic excitations carrying no fractional quantum number under the ordinary symmetry. To reach the latter conclusion, we have used the precise form the anomaly, instead of just the presence of the anomaly. As the referee suggested, we have expanded Section VI in the revised manuscript.

Report:

This paper explores the 't Hooft anomalies of ordinary and invertible finite 1-form symmetries in the Kitaev spin-$S$ model. The authors diagnose the 't Hooft anomaly using a technique common in the study of topological order, showing that the anomaly is trivial for $S\in\mathbb{Z}$, while nontrivial for $S\in\mathbb{Z}+\frac{1}{2}$. As they state in their paper, this implies that the Kitaev spin-S model, and any symmetric deformed versions, with $S\in\mathbb{Z}+\frac{1}{2}$ cannot realize a trivial phase. The paper is nicely written and has the benefit of discussing ideas common throughout the generalized symmetry literature in a way cond-mat physics folks outside this community should be able to follow and learn from. For this reason, the paper should ultimately be published. However, I would first like to see some minor revisions before publication. As I explain below, a few places in the main text contain minor errors (or perhaps need clarification). Furthermore, while I do not require related revisions, the paper lacks novelty at times, and I would love to see the authors expand on section VI (see suggestions below).

Response:

We thank the referee for the positive feedback of our work, as well as the constructive comments, suggestions and questions, which are helpful for us to improve the paper. The comments and questions are addressed below.

Report:

Some general comments/questions: (1) It would be helpful for the reader if the authors could summarize the results in Refs. 33-35 in more detail to contextualize their results with the established literature and emphasize the novelty of their work. I imagine one key result absent from the literature is the anomalies' dependency on $S$. If this is true, it would be helpful for the authors to state it clearly. One way the authors could improve the novelty of their paper is to include the lattice symmetries and study their interplay with the internal symmetries, which is currently stated as something left to future work. This interplay would be fascinating and be mathematically described by a nontrivial 2-group. This interplay would be related to what Kitaev called ``weak symmetry breaking" in Appendix F of [1], and the group homomorphism $\omega_1: G\rightarrow\Gamma_1$ in Appendix F would be the defining data for the 2-group. Similar 2-group symmetries involving lattice symmetries were discussed in arXiv:2301.04706.

Response:

As far as we understand, this comment is regarding the new results in the present paper compared to Refs. 33-35. However, as discussed in the Introduction, the motivation of the present paper starts from the study of Kitaev magnetism, and the context is set by Refs. 1-28, instead of Refs. 33-35. Furthermore, as summarized in the abstract and Introduction, and as discussed in more detail in the paper, here we discuss not only the anomaly associated with the 1-form symmetry, but also the anomaly related to the 0-form internal symmetry, and our discussion covers all $S$. From this analysis, we conclude the existence of deconfined fermionic excitations with no fractional quantum number under the 0-form symmetry for all $S\in\mathbb{Z}+\frac{1}{2}$. Refs. 33-35 contain many results not relevant to the present paper, and the results related to the present paper, namely, the anomaly of the 1-form symmetry for some specific $S$, is just one of the many components of this paper. So we think it is reasonable to just cite these papers in Sec. III. The discussion about lattice symmetries is indeed interesting, but we consider it to be beyond the scope of the present work.\

Report:

(2) It would be interesting if the authors could comment on modifications of the Kitaev spin-S model that would change the 't Hooft anomaly. For instance, modifications that would give rise to symmetry fractionalization patterns. For example, what I have in mind is something like appendix A of arXiv:2206.14197, which does this for the toric code and translation symmetry. It would be neat to see a similar exploration of the Kitaev spin-S model, which I believe lies in the scope of the paper and may be doable without importing fancy numerics techniques.

Response:

In both the main text and in more detail in the Appendix, we showed that the anomaly of the 1-form symmetry is reflected in the algebra of operators that define the 1-form symmetry. So to change the anomaly, say, of the 1-form symmetry, one simply needs to find another set of operators that satisfy another set of algebra, and find a Hamiltonian with such a symmetry. However, we think such a discussion would not be directly related to any Kitaev material.

Report:

(3) It would be interesting if the authors could discuss the corresponding 3+1D SPT for their 't Hooft anomaly. I would imagine that the topological response theory is the well-known action $S=i\pi\int A\cup A$, where $A$ is a $\mathbb{Z}_2$ 2-cocycle. But it would be interesting to see this explored on the lattice by finding a $3+1$D lattice Hamiltonian of the SPT whose boundary is the Kitaev spin-S model. Since the modern perspective of both the cond-mat and hep-th communities on 't Hooft anomalies of invertible symmetries is through its corresponding SPT, this feels like a relevant addition to the paper.

Response:

Indeed, in the Appendix we write down the same action, albeit in a different notation. However, since the motivation of the present work starts from the study of Kitaev magnetism, diving into the details of constructing a $3+1$D lattice model that realizes the SPT corresponding to this anomaly may drive the focus of the readers away.

Report:

(4) As mentioned in the above weakness section, section VI is very brief. The authors correctly state the beautiful consequence of 't Hooft anomalies preventing the model from having a trivial phase, and they nicely contextualize this with known phases. I would love to see the authors explore interesting $\mathbb{Z}_2^{[1]}$ symmetric deformations of their model to investigate new interesting phases and further explore the hypothesis of emergibility. Another suggestion for the authors to expand section VI is to discuss how the microscopic anomalous $\mathbb{Z}_2^{[1]}$ symmetry is related to the IR symmetries in the different phases (particularly the $\mathbb{Z}_2$ and Ising topologically ordered phases). Some of these suggestions are mentioned in the paper's discussion section, but I feel it is a real shame not to explore some of them in the present paper.

**Response: **

In the revised manuscript, we have added to the list of possible phases compatible with the symmetry and anomaly. However, to pin down which microscopic models give rise to these phases requires extensive numerical studies, which is beyond the scope of this paper. The relation between the microscopic anomalous $\mathbb{Z}_2^{[1]}$ symmetry and the IR symmetries of the different phases is implicitly discussed in the original manuscript, and in the revised version we have made it more explicit.

Report:

Some specific comments/ questions: (1) In paragraph 2 of the introduction, the authors write, ``The $\mathbb{Z}_2$ gauge field is deconfined if $S\in\mathbb{Z}+\frac{1}{2}$, but it can be Higgsed if $S\in\mathbb{Z}$ [28]." I am confused about the wording here. Are they saying that for $S\in\mathbb{Z}+\frac{1}{2}$, the model cannot undergo a Higgs transition into a trivial phase and is always in a deconfined phase regardless of perturbations added, while for $S\in\mathbb{Z}$ the model has both a deconfined and Higgs phase? I am also confused about what is being Higgsed here since the deconfined topological excitation discussed later on is a fermion. It would be helpful if the authors could reword this sentence to ensure their message is clearly communicated.

**Response: **

The understanding of the referee is correct. If $S\in\mathbb{Z}+\frac{1}{2}$, the charge of the $\mathbb{Z}_2$ gauge field is fermionic, so it indeed cannot undergo a Higgs transition. On the other hand, if $S\in\mathbb{Z}$, the charge is bosonic, and the model can indeed have both a deconfined and Higgs phase. We have modified this sentence in the revised manuscript.\

Report:

(2) It would be helpful for the reader if the authors could cite references [31, 46–48] after writing, ``Moreover, even if the 1-form symmetry is slightly broken by local perturbations, its consequences are still robust" in the fifth paragraph of the introduction. Furthermore, there are a handful of other papers that should be cited alongside [31, 46–48] that make similar claims from different perspectives (e.g., arXiv:2106.12610, arXiv:2211.12543, arXiv:2202.05866, arXiv:2304.13751).

**Response: **

The readers may be confused if we cite the previous literature while summarizing the main results of this paper. These references are cited in Sec. VI, where the details of corresponding physics are explicitly discussed. We have also cited the additional papers suggested by the referee there.\

Report:

(3) When introducing Eq 2, The authors really should cite [1] since Kitaev wrote down this loop operator for the S=1/2 case (Eq. 6 of [1] with $\sigma^a=-ie^{i\pi S^a}$ as well as [33] where the same operator $W_p$ is written down in the first paragraph of Sec. IIIC.

**Reponse: **

We have explicitly cited the papers where this operator was first introduced in the revised manuscript.

**Report: **

(4) The authors define 't Hooft anomaly in section III as any obstruction to gauging. They may want to be more specific in their definition since not all obstructions to gaugings are 't Hooft anomalies (e.g., one cannot gauge the $\mathbb{Z}_4/\mathbb{Z}_2$ part of a $\mathbb{Z}_4$ symmetry, which may seem silly to try but is a simplified example of obstructions to gauging parts of anomaly-free higher-group symmetries). In the context of invertible symmetries, 't Hooft anomalies are only obstructions to gauging when they can remedied by an SPT in one higher dimension.

Reponse:

In our context, it should be clear that the symmetry we consider is the $\mathbb{Z}_2$ 1-form symmetry and no confusion is likely to arise. We think it makes sense to state the anomaly in the way as it is.\

Report:

(5) In the final paragraph of section III, the authors state that the anomaly can also be seen from the anisotropic limit. The calculation leading up to this paragraph is lovely and proves there is indeed an anomaly. I do agree that the fact that the anisotropic limit for $S\in\mathbb{Z}$ is trivial implies there is no anomaly, but I do not agree that just because it is nontrivial for $S\in\mathbb{Z}+\frac{1}{2}$ that it means there is an anomaly. The latter observation is consistent with a 't Hooft anomaly and a sound check, but I don't think it proves there has to be one. Could the authors elaborate on this?

**Response: **

The reasoning for the absence of anomaly when $S\in\mathbb{Z}$ is indeed that a trivial phase can be realized in that case. However, as stated in the manuscript, the anomaly for $S\in\mathbb{Z}+\frac{1}{2}$ is seen from 1) the existence of a $\mathbb{Z}_2$ topological order and 2) the closed loop operators discussed in this paper is precisely the Wilson loop of the deconfined fermions in the $\mathbb{Z}_2$ topological order.

Report:

(6) In the second paragraph of section VI, the authors write, ``...the anomaly of the $\mathbb{Z}_2^{[1]}$ symmetry implies that at low energies there are deconfined fermionic excitations..." Does this mean the authors claim that the 't Hooft anomaly can only be saturated by the $\mathbb{Z}_2^{[1]}$ symmetry spontaneously breaking? The claim is somewhat reasonable, but I don't see why it has to be true. Why is it not possible, for example, to add symmetric terms to the Kitaev spin-S model that can drive a phase transition to a gapless phase where the $\mathbb{Z}_2^{[1]}$ symmetry is not spontaneously broken?

**Reponse: **

We made no claim regarding whether the $\mathbb{Z}_2^{[1]}$ symmetry is spontaneously broken or not. The claim about the deconfined fermionic excitations, in the field theoretic language, means that the low-energy effective theory has a one dimensional topological defect with topological spin $-1$. We have added a sentence regarding this point in the revised manuscript.

Report:

(7) In the first paragraph of Appendix Section 3, the authors comment on the onsite aspect of the anomalous 1-form symmetry. I want to remark that the onsite versus non-onsite diagnosis for Hooft anomalies of generalized symmetries fails pretty often. For instance, In 1+1D Hamiltonian G gauge theory, where G is a finite group, the $Rep(G)$ 0-form symmetry is anomaly-free but not always onsite (the symmetry operators are all matrix product operators). It is pretty standard that non-invertible symmetries of 1+1D Hamiltonian models are matrix product operators and are not on site (e.g., see arXiv:2112.09091).

**Response: **

We agree with the referee that non-onsiteness of a symmetry does not imply the symmetry is anomalous. On the other hand, for invertible 0-form internal symmetries, the existence of an on-site implementation of a symmetry in a tensor-product Hilbert space means that it is anomaly free. We expand on this point in the paragraph.

Report:

(8) There is a typo in Eq. B1: $B\mathbb{Z}_2^{y)}$ should be $B\mathbb{Z}_2^{y)}$. Also, it may be helpful for the mathematically inclined reader to explain that $B\mathbb{Z}_2^{(x)}\times B\mathbb{Z}_2^{(x)}\times B^2\mathbb{Z}_2$ in Eq. B1 started its life as the classifying space of a 2-group $G$ related to the split fibration $K(\mathbb{Z}_2, 2)\rightarrow B\mathbb{G}\rightarrow K(\mathbb{Z}_2^{(x)}\times\mathbb{Z}_2^{(y)}, 1)$ in a Postnikov system.

Response:

We thank the referee for carefully reading our paper and pointing out this typo. We also thank the referee for the suggestion regarding 2-group. We have corrected the typo and added a footnote regarding 2-group.

---

## Round 1 · Referee Report · Anonymous (Referee 2) · 2024-1-30

Report

This paper investigates the Kitaev spin-S models by a thorough analysis of the global symmetry and the corresponding anomaly.

First, the authors list all the global symmetries of the Kitaev spin-S models, including the one form $\mathbb{Z}_2^{[1]}$ symmetry, and study the classification of anomaly corresponding to all global symmetries. Then, they pin down the anomaly for spin-1/2 model, which only comes from the one form$ \mathbb{Z}_2^{[1]}$ symmetry. The author further find the anomaly for general spin-S model, which gives nontrivial constraints on the ground state of these models.

This constraints from the anomaly will give new insight on the further study of the ground state of Kitaev spin-S models. The draft is clearly articulated and I recommend to publish this draft in SciPost.

Requested changes

  1. Can you provide a short explanation on why $H_2$ in equation (7) is the effective Hamiltonian of the Kitaev spin-S model?

  2. There are several typos that need to be changed: 1). In the second term of $H_2$ in equation (7), the label for $s^{\mu}_{\beta i}$ should be $s^{\mu}_{\beta j}$. 2). In equation B1, the label for $B\mathbb{Z}_2^{y)}$ should be $B\mathbb{Z}_2^{(y)}$

  • validity: -
  • significance: -
  • originality: -
  • clarity: -
  • formatting: -
  • grammar: -

Author:  Liujun Zou  on 2024-03-10  [id 4355]

(in reply to Report 2 on 2024-01-30)

We thank the referee for the report. The question and requested changes are helpful for us to improve the paper, and they are addressed below.

Report:

This paper investigates the Kitaev spin-S models by a thorough analysis of the global symmetry and the corresponding anomaly. First, the authors list all the global symmetries of the Kitaev spin-S models, including the one form $\mathbb{Z}^{[1]}_2$ symmetry, and study the classification of anomaly corresponding to all global symmetries. Then, they pin down the anomaly for spin-1/2 model, which only comes from the one form $\mathbb{Z}^{[1]}_2$ symmetry. The authors further find the anomaly for general spin-S model, which gives nontrivial constraints on the ground state of these models. This constraints from the anomaly will give new insight on the further study of the ground state of Kitaev spin-S models. The draft is clearly articulated and I recommend to publish this draft in SciPost.

Response:

We thank the referee for recommending our work for publication.

Report:

  1. Can you provide a short explanation on why $H_2$ in equation (7) is the effective Hamiltonian of the Kitaev spin-S model?

Response:

When $J_2\gg|J_3^{x,y,z}|$, we can first ignore $J_3^{x,y,z}$ and consider only $J_2$. The $J_2$ term forces all $2S$ species of spin-1/2's to form a spin-$S$ moment. Within the low-energy Hilbert space made of these spin-$S$ moments, the $J_3^{x,y,z}$ term precisely gives the Kitaev model. We have added a footnote explaining this in the revised manuscript.

Report:

  1. There are several typos that need to be changed: 1). In the second term of $H_2$ in equation (7), the label for $s^\mu_{\beta i}$ should be $s^\mu_{\beta j}$. 2). In equation B1, the label for $B\mathbb{Z}_2^{y)}$ should be $B\mathbb{Z}_2^{(y)}$.

Response:

We thank the referee for carefully reading our paper and pointing out these typos. We have corrected these typos.

---

## Round 1 · Referee Report · Anonymous (Referee 3) · 2024-2-20

Strengths

  1. This paper clearly articulates a goal of extending the results of Ref. [28], which identified Z_2 spin liquid behavior in a non-integrable spin-S version of the Kitaev honeycomb lattice model.

  2. It carefully explores the symmetries and anomalies of the spin-S model, and the distinctions between integer and half-odd-integer S.

  3. It is in general clearly written, and the analytical arguments are all laid out in good detail.

  4. It establishes these results based on the algebra of the spin operators themselves, without relying on the parton construction of Ref. [28].

Weaknesses

I detect no significant weaknesses.

Report

This appears to be a high-quality work, and the writing is excellent. I am happy to recommend publication in SciPost.

Requested changes

I am not requesting any changes, but I do have some questions which might possibly be worthy of consideration. (They may also reflect my own confusion.)

First, is there an analog of Lieb's theorem which constrains the ground state plaquette fluxes in Eqn. (2)? If so, it doesn't seem obvious, since in Ma's parton construction the Hamiltonian is not equivalent to a single species of Majorana fermion hopping in a static Z_2 background gauge field.

Second, In Eqn. (B8) the authors introduce a three-spin term which breaks time-reversal symmetry - the same one considered by Kitaev in his S=1/2 model. Another route to breaking T is by introducing odd-membered plaquettes, as in the work of Yao and Kivelson on the decorated honeycomb lattice, where each vertex of the original honeycomb becomes a triangle, resulting in 12-sided and 3-sided loops. The flux of odd-membered loops is odd under time-reversal. Are such considerations also relevant to the spin-S model on the decorated honeycomb lattice?

  • validity: high
  • significance: good
  • originality: good
  • clarity: high
  • formatting: excellent
  • grammar: perfect

Author:  Liujun Zou  on 2024-03-10  [id 4356]

(in reply to Report 3 on 2024-02-20)

We thank the referee for the report, which is helpful for us to improve the paper. The questions are addressed below.

Report:

Strengths: 1. This paper clearly articulates a goal of extending the results of Ref. [28], which identified $Z_2$ spin liquid behavior in a non-integrable spin-S version of the Kitaev honeycomb lattice model. 2. It carefully explores the symmetries and anomalies of the spin-S model, and the distinctions between integer and half-odd-integer $S$. 3. It is in general clearly written, and the analytical arguments are all laid out in good detail. 4. It establishes these results based on the algebra of the spin operators themselves, without relying on the parton construction of Ref. [28]. Weaknesses: I detect no significant weaknesses. Report: This appears to be a high-quality work, and the writing is excellent. I am happy to recommend publication in SciPost.

Response:

We thank the referee for thinking highly of our work.

Report:

Requested changes: I am not requesting any changes, but I do have some questions which might possibly be worthy of consideration. (They may also reflect my own confusion.) First, is there an analog of Lieb's theorem which constrains the ground state plaquette fluxes in Eqn. (2)? If so, it doesn't seem obvious, since in Ma's parton construction the Hamiltonian is not equivalent to a single species of Majorana fermion hopping in a static $Z_2$ background gauge field.

Response:

We cannot prove a Lieb's theorem in the current context. On the other hand, as mentioned in footnote 1 and the beginning of Appendix A 1, whether the ground states are in the zero-flux sector does not affect our discussion, and they can be in a generic flux sector. On the other hand, it is indeed important that the flux-flipping excitations, which are excitations that are in a different flux sector as the ground states, are gapped. We have added a paragraph in the revised manuscript to emphasize this point.

Report:

Second, In Eqn. (B8) the authors introduce a three-spin term which breaks time-reversal symmetry - the same one considered by Kitaev in his S=1/2 model. Another route to breaking T is by introducing odd-membered plaquettes, as in the work of Yao and Kivelson on the decorated honeycomb lattice, where each vertex of the original honeycomb becomes a triangle, resulting in 12-sided and 3-sided loops. The flux of odd-membered loops is odd under time-reversal. Are such considerations also relevant to the spin-S model on the decorated honeycomb lattice?

Response:

We believe that the referee is discussing the work by Yao and Kivelson in Phys. Rev. Lett. 99, 247203. In the spin-$S$ version of that model, one can define 3-sided loops as $ie^{i\pi(S_1^x+S_2^y+S_3^z)}$ if $S\in\mathbb{Z}+\frac{1}{2}$, where the subscripts $1,2,3$ refer to the 3 sites around a small triangle. This loop operator is odd under time reversal and Hermitian, so it can be added into the Hamiltonian to explicitly break the time reversal symmetry. However, if $S\in\mathbb{Z}$, the Hermitian 3-sided loop is $e^{i\pi(S_1^x+S_2^y+S_3^z)}$, which is even under time reversal, so adding it to the Hamiltonian does not break the time reversal symmetry, although $ie^{i\pi(S_1^x+S_2^y+S_3^z)}$ can still be used as an order parameter for the time reversal symmetry.

---

## Round 2 · Author Response

We thank the Editor for handling our manuscript and Referees for their reports, which are helpful for us to improve our paper. We address the comments and questions of the referees in the submission page of this paper. We also summarize the changes of the paper below.

---

## Round 2 · List of Changes

We made various changes listed below, most of which are to address the comments and questions of the referees.

  1. We have changed the title to add the term ``symmetry-enforced exotic quantum matter" to extend the scope of the paper. This phrase is also added in the abstract, introduction, main text and discussion section of the paper.

  2. We have restructured the paper, so that the section about the spin-1/2 case and section on the even-odd effect are now combined into a single section.

  3. We have expanded Sec. VI to discuss the consequences of the symmetries and anomalies. In particular, we have enumerated more quantum phases compatible with the anomalies that are not known within the Kitaev spin-1/2 model.

  4. We have added a sentence to explain the meaning of ``deconfined fermionic excitations" in the language of topological line defects in quantum field theory.

  5. We have added a footnote to further explain the interpolation leading to the even-odd effect.

  6. We have added a paragraph to explain that our results rely on the assumption that the energy gap to flip the eigenvalue of the $W_p$ operator is finite. This condition holds generically unless the Hamiltonian is fine tuned, so our results are valid for almost all Hamiltonians with the relevant symmetries and anomalies.

  7. We have corrected various typos and added some new references.

---

## Editorial Decision

published